# Synthetic data shuffling accelerates the convergence of federated learning under data heterogeneity

## Abstract

In federated learning, data heterogeneity is a critical challenge. A straightforward solution is to shuffle the clients' data to homogenize the distribution. However, this may violate data access rights, and how and when shuffling can accelerate the convergence of a federated optimization algorithm is not theoretically well understood. In this paper, we establish a precise and quantifiable correspondence between data heterogeneity and parameters in the convergence rate when a fraction of data is shuffled across clients. We discuss that shuffling can in some cases quadratically reduce the gradient dissimilarity with respect to the shuffling percentage, accelerating convergence. Inspired by the theory, we propose a practical approach that addresses the data access rights issue by shuffling locally generated synthetic data. The experimental results show that shuffling synthetic data improves the performance of multiple existing federated learning algorithms by a large margin.

## 1 Introduction

Federated learning (FL) is emerging as a fundamental distributed learning paradigm that allows a central server model to be learned collaboratively from distributed clients without ever requesting the client data, thereby dealing with the issue of data access rights (Konečný et al., 2016; Kairouz et al., 2019; Wang et al., 2021; Sheller et al., 2020). One of the most popular algorithms in FL is FedAvg (Konečný et al., 2016): A server distributes a model to the participating clients, who then update it with their local data for multiple steps before communicating it to the server, where the received client models are aggregated, finishing one round of communication. Two main challenges in FedAvg optimization are 1) large data heterogeneity across clients and 2) limited available data on each client (Konečný et al., 2016).

Data heterogeneity refers to differences in data distributions among participating clients (Kairouz et al., 2019). Existing works typically characterize its impact on the convergence of FedAvg using bounds on the gradient dissimilarity, which suggests a slow convergence when the data heterogeneity is high (Karimireddy et al., 2020b; Koloskova et al., 2020). Many efforts have been made to improve FedAvg's performance in this setting using advanced techniques such as control variates (Karimireddy et al., 2020b; Acar et al., 2021; Li et al., 2022a) and regularizers (Wang et al., 2020). Although these algorithms have demonstrated great success in many applications, they are insufficient under high data heterogeneity (Yu et al., 2022). On the other hand, many researchers (Zhao et al., 2018; Woodworth et al., 2020) have observed that simply augmenting the dataset on each client with a small portion of shuffled data collected from all clients can significantly accelerate the convergence of FedAvg. However, it has yet to be well understood when and, in particular, by how much shuffling can accelerate the convergence.

Another challenge arises when the amount of available data on each client is limited, especially when the number of classes or tasks is large. High-capacity deep neural networks (DNN) usually require large-scale labelled datasets to avoid overfitting (He et al., 2016; Shrivastava et al., 2016). However, labelled data acquisition can be expensive and time-consuming (Shrivastava et al., 2016). Therefore, learning with conditionally generated synthetic samples has been an active area of research (Sehwag et al., 2022; Goetz & Tewari, 2020; Shrivastava et al., 2016; Zhang et al., 2022; Zhu et al., 2021; Li

Figure 1: Our proposed framework. (a) Each client learns a generator with a subset of its local data and generates synthetic data, which are communicated to the server. The server then shuffles and sends the partitioned collection of the synthetic data to each client. (b) With the updated local data, any FL algorithms can be used to learn a server model. (c) When the clients are very heterogeneous, compared to shuffling the real data, shuffling synthetic data achieves a similar accuracy while alleviating information leakage.

et al., 2022b). Adapting the framework of using synthetic data from centralized learning to FL is not trivial due to the unique properties of FL, such as distributed datasets, high communication costs, and privacy requirements (Kairouz et al., 2019).

In this work, we rigorously study the correspondence between the data heterogeneity and the parameters in the convergence rate via shuffling. Specifically, we show that reducing data heterogeneity by shuffling in a small percentage of data from existing clients can give a quadratic reduction in the gradient dissimilarity in some scenarios and can lead to a super-linear reduction (a factor greater than the shuffle percentage) in the number of rounds to reach the target accuracy. Further, we show that it holds in both strongly-convex functions and non-convex DNN-based FL.

While we theoretically understand the optimization improvement from shuffling, gathering real data from clients goes against the goal of protecting privacy in FL. Therefore, we propose a more practical framework `Fedssyn` where we shuffle a collection of synthetic data from all the clients at the beginning of FL (see Fig. 1). Each client learns a client-specific generator with a subset of its local data and generates synthetic data. The server then receives, shuffles, partitions, and redistributes the synthetic data to each client. Collecting visually interpretable synthetic data can be more transparent and secure than collecting the black-box generators. This transforms the issue into one concerning the assurance of privacy in the generative model, an area of active research that already boasts some well-established techniques Yoon et al. (2019); Chen et al. (2020); Wang et al. (2023).

**Contributions:** We summarize our main results below:

- We rigorously decompose the impact of shuffling on the convergence rate in FedAvg. Our careful discussion provides insights in understanding when and how shuffling can accelerate the convergence of FedAvg. We empirically verify our theoretical statements on strongly convex and DNN-based non-convex functions.

- Inspired by the theoretical understanding of the effects of data shuffling, we present a practical framework Fedssyn, that shuffles for privacy reasons locally generated *synthetic data* and can be coupled with *any* FL algorithms.

- We empirically demonstrate that using `Fedssyn` on top of several popular FL algorithms reduces communication cost and improves the Top-1 accuracy. Results hold across multiple datasets, different levels of data heterogeneity, number of clients, and participation rates.

## 1.1 RELATED WORK

**Federated optimization:** FL is a fast-growing field (Wang et al., 2021). We here mainly focus on works that address the optimization problems in FL. FedAvg Konečný et al. (2016) is one of the most commonly used optimization techniques. Despite its success in many applications, its convergence performance under heterogeneous data is still an active area of research (Karimireddy et al., 2020b; Wang et al., 2022a; Khaled et al., 2019; Sahu et al., 2018; Woodworth et al., 2020; Li et al., 2022a). Theoretical results suggest that the *drift* caused by the data heterogeneity has a strong negative impact on FedAvg convergence. Several lines of work have been proposed to reduce

the impact of data heterogeneity, including regularizers (Sahu et al., 2018), control variates for correcting gradients (Karimireddy et al., 2020b; Acar et al., 2021; Li et al., 2022a; Mitra et al., 2021), and decoupled representation and classification (Collins et al., 2022; Luo et al., 2021; Yu et al., 2022), sophisticated aggregation strategies (Lin et al., 2020; Karimireddy et al., 2020a; Wang et al., 2020; Reddi et al., 2020). However, such methods fall short in addressing high client heterogeneity and fail to achieve accuracies on par with those in the IID setting (Yu et al., 2022)

**Synthetic data-based FL:** Many works attempt to improve the performance of FL algorithms using synthetic data (Zhu et al., 2021; Goetz & Tewari, 2020). We can group them into 1) client-generator (Xiong et al., 2023; Li et al., 2022b; Wang et al., 2023), where a generator is trained locally and sent to the server for synthetic data generation, and 2) server-generator (Zhu et al., 2021; Zhang et al., 2022), where a centralized generator helps with updating the server and/or client model. Comparably, our framework has removed many complicated components as shown in Table A.1.

**Generative models:** Generative models can be trained to mimic the underlying distribution of complex data modalities (Goodfellow et al., 2014). Generative Adversary Network (GANs) have achieved great success in generating high-quality images (Sinha et al., 2021; Zhao et al., 2020; Karras et al., 2020); however, training GANs in a FL setup is challenging as each client only has limited data (Salimans et al., 2016; Konečný et al., 2016). Diffusion models (Ho et al., 2020) have attracted much attention due to the high diversity of the generated samples Sehwag et al. (2022) and training efficiency (Ho et al., 2020). Therefore, we study to use DDPM as our local generator in this paper.

**Differentially private learning in neural networks:** Differentially private stochastic gradient descent (DP-SGD) bounds the influence of any single input on the output of machine learning algorithms to preserve privacy (McMahan et al., 2018; Shokri & Shmatikov, 2015; Abadi et al., 2016; Xie et al., 2018; Chen et al., 2020; Yoon et al., 2019). However, this paper aims to analyze the impact of shuffling on the convergence rate from the optimization perspective. Using both DP-generator and DP-FL algorithms complicates the problem as it introduces complicated trade-off between quality of synthetic data, model utility, and privacy budget. Therefore, we leave this as our future work.

## 2   INFLUENCE OF THE DATA HETEROGENEITY ON THE CONVERGENCE RATE

We formalize the problem as minimizing the expectation of a sum of stochastic functions $F$ (e.g. the loss associated with each datapoint):

$$f^\star := \min_{\mathbf{x} \in \mathbb{R}^d} \left[ \frac{1}{N} \sum_{i=1}^{N} f(\mathbf{x}, \mathcal{D}_i) \right], \quad f(\mathbf{x}, \mathcal{D}_i) := \mathbb{E}_{\xi \sim \mathcal{D}_i} F(\mathbf{x}, \xi),$$

where $\mathcal{D}_i$ represents the distribution of $\xi$ on client $i$ and we use $\mathcal{D} := \bigcup_i \mathcal{D}_i$ to denote the uniform distribution over the joint data. We assume gradient dissimilarity, bounded noise, and smoothness of the function, following Koloskova et al. (2020); Stich (2019):

**Assumption 1** (gradient dissimilarity). *We assume that there exists $\zeta^2 \geq 0$ such that $\forall \mathbf{x} \in \mathbb{R}^d$:*

$$\frac{1}{N} \sum_{i=1}^{N} ||\nabla f(\mathbf{x}, \mathcal{D}_i) - \nabla f(\mathbf{x}, \mathcal{D})||^2 \leq \zeta^2.$$

**Assumption 2** (stochastic noise). *We assume that there exist $\sigma_{avg}^2 \geq 0$ and $\sigma^2 \geq 0$ such that $\forall \mathbf{x} \in \mathbb{R}^d$, $i \in [N]$:*

$$\mathbb{E}_{\xi \sim \mathcal{D}_i} ||\nabla F(\mathbf{x}, \xi) - \nabla f(\mathbf{x}, \mathcal{D}_i)||^2 \leq \sigma^2, \quad \mathbb{E}_{\xi \sim \mathcal{D}} ||\nabla F(\mathbf{x}, \xi) - \nabla f(\mathbf{x}, \mathcal{D})||^2 \leq \sigma_{avg}^2.$$

**Assumption 3** (smoothness). *We assume that each $F(\mathbf{x}, \xi)$ for $\xi \in \mathcal{D}_i$, $i \in [N]$ is L-smooth, i.e.,:*

$$||\nabla f(\mathbf{x}, \mathcal{D}_i) - \nabla f(\mathbf{y}, \mathcal{D}_i)|| \leq L||\mathbf{x} - \mathbf{y}||, \qquad \forall \mathbf{x}, \mathbf{y} \in \mathbb{R}^d.$$

These assumptions are standard in the literature (cf. Koloskova et al., 2020; Woodworth et al., 2020; Khaled et al., 2020). We further denote by $L_{\text{avg}} \leq L$ the smoothness of $f(\mathbf{x}, \mathcal{D})$, and by $L_{\text{max}} \leq L$ a uniform upper bound on the smoothness of each single $f(\mathbf{x}, \mathcal{D}_i)$, for all $i \in [N]$.

### 2.1   MOTIVATION

Data heterogeneity is unavoidable in real-world FL applications since each client often collects data individually (Konečný et al., 2016). One simple but effective strategy to address data heterogeneity

is to replace a small fraction (e.g. 10%) of the client's data with shuffled data collected from other clients. Woodworth et al. (2020); Zhao et al. (2018) have observed that shuffling can vastly reduce the reached error. In the following, we focus on explaining this empirically tremendous optimization improvement from a theoretical perspective. However, we emphasize that this is not a practical approach, as data sharing may be diametrically at odds with the fundamentals of FL.

## 2.2 MODELLING DATA SHUFFLING AND THEORETICAL ANALYSIS

We formally describe the data heterogeneity scenario as follows: We assume each client has access to data with distribution $\mathcal{D}_i$ that is non-iid across clients and we denote by $\mathcal{D} := \bigcup_i \mathcal{D}_i$ the uniform distribution over the joint data. We can now model our shuffling scenario where the server mixes and redistributes a small $p$ fraction of the data by introducing new client distributions $(1-p)\mathcal{D}_i + p\mathcal{D}$ for a shuffling parameter $p \in [0, 1]$. If $p = 0$, the local data remains unchanged, and if $p = 1$, the data is uniform across clients.

We argued that in practice it might be infeasible to assume access to the distribution $\mathcal{D}$. Instead, only an approximation (e.g. synthetic data) might be available, which we denote by $\tilde{\mathcal{D}}$. Accordingly, we define $\tilde{\sigma}_{\text{avg}}$ similarly as Assumption 2 but with $\xi \sim \tilde{\mathcal{D}}$ instead of $\xi \sim \mathcal{D}$ and define $\tilde{L}_{\text{avg}}$ similarly as $L_{\text{avg}}$ but on $\tilde{\mathcal{D}}$. We denote the client distribution after shuffling by $\hat{\mathcal{D}}_i := (1-p)\mathcal{D}_i + p\tilde{\mathcal{D}}$ and we quantify the data distribution differences by a new parameter $\delta$. Note that in an idealized scenario when $\tilde{\mathcal{D}} = \mathcal{D}$, then $\delta = 0$. See Appendix A.1 for more details.

**Assumption 4** (distribution shift). *We assume that there exists $\delta^2 \geq 0$ such that $\forall \mathbf{x} \in \mathbb{R}^d$:*

$$||\nabla f(\mathbf{x}, \mathcal{D}) - \nabla f(\mathbf{x}, \tilde{\mathcal{D}})||^2 \leq \delta^2 \,.$$

We now characterize how shuffling a $p$ fraction of data to each client impacts the problem difficulty. For clarity, we will use the index $p$ to denote how stochastic noise $\hat{\sigma}_p^2$, gradient dissimilarity $\hat{\zeta}_p^2$ and the smoothness constant $L_p$ can be estimated for the new distributions $\hat{\mathcal{D}}_i$ resulting after shuffling.

**Lemma 1.** *If Assumption 1 – 4 hold, then in expectation over potential randomness in selecting $\tilde{\mathcal{D}}$:*

$$\mathbb{E}\hat{\sigma}_p^2 \leq (1-p)\sigma^2 + p\tilde{\sigma}_{avg}^2 + p(1-p)\zeta^2 + p(1-p)\delta^2 \,, \qquad \mathbb{E}\hat{\zeta}_p^2 \leq (1-p)^2\zeta^2 \,,$$

$$\mathbb{E}L_p \leq (1-p)L_{max} + p\tilde{L}_{avg} \,.$$

Lemma 1 shows the effective stochastic noise, gradient dissimilarity, and function smoothness when each client $i$ contains a fraction $p$ of shuffled data (see Appendix A.1 for the proof). We observe that the gradient dissimilarity decays quadratically with respect to $p$. When $p = 1$, the gradient dissimilarity is 0 as expected, since data is iid across workers. We also characterize the impact of the distribution shift from $\mathcal{D}$ to $\tilde{\mathcal{D}}$ on the effective stochastic noise. We now demonstrate the impact of these parameters on the convergence by utilizing the convergence bounds established in Koloskova et al. (2020); Woodworth et al. (2020); Khaled et al. (2020) and combining them with our Lemma 1:

**Corollary I** (Convergence rate after shuffling). *We consider: $p$ as the fraction of shuffled data on each client; $\tau$ as local steps; under Assumption 1– 4, for any target accuracy $\varepsilon > 0$, there exists a (constant) stepsize such that the accuracy can be reached after at most $T$ iterations (in expectation):*

$$\textbf{Strongly-convex} : T = \mathcal{O}\left( \frac{\hat{\sigma}_p^2}{\mu N \varepsilon} + \frac{\sqrt{L}\left(\tau\hat{\zeta}_p + \sqrt{\tau}\hat{\sigma}_p\right)}{\mu\sqrt{\varepsilon}} + \frac{L\tau}{\mu} \log\frac{1}{\varepsilon} \right) \,,$$

$$\textbf{Non-convex} : T = \mathcal{O}\left( \frac{L\hat{\sigma}_p^2}{N\varepsilon^2} + \frac{L\left(\tau\hat{\zeta}_p + \sqrt{\tau}\hat{\sigma}_p\right)}{\varepsilon^{3/2}} + \frac{L\tau}{\varepsilon} \right) \,.$$

From Corollary I, we see that if $\hat{\sigma}_p^2 > 0$, the convergence rate is asymptotically dominated by the first term, i.e. $\mathcal{O}\left(\frac{\hat{\sigma}_p^2}{\mu N \varepsilon}\right)$, which can be bounded by $\mathcal{O}\left(\frac{(1-p)\sigma^2 + p\tilde{\sigma}_{\text{avg}}^2 + p(1-p)\zeta^2 + p(1-p)\delta^2}{N\mu\varepsilon}\right)$. This shows a mixed effect between the stochastic noise, gradient dissimilarity, and the distribution shift. Shuffling can have less of an impact on the convergence in the high noise regime. When $\sigma^2 = 0$, or in general in the early phases of the training (when the target accuracy $\varepsilon$ is not too small), for problems where Lemma 1 holds, then the number of iterations $T$ to achieve accuracy $\varepsilon$ can be super-linearly reduced by a factor larger than $p$ with the increasing ratio of the shuffled data $p$, as $T$ is proportional to $\hat{\zeta}_p$ and $\sqrt{L}_p$ in expectation for strongly convex functions. We next present a numerical example to verify this.

### 2.3 ILLUSTRATIVE EXPERIMENTS ON CONVEX FUNCTIONS

We here formulate a quadratic example to verify that our theoretical results when we shuffle a $p$ faction of the local data. Following[1] Koloskova et al. (2020), we consider a distributed least squares objective $f(\mathbf{x}) := \frac{1}{n}\sum_{i=1}^{n}\left[f_i(\mathbf{x}) := \frac{1}{2n_i}\sum_{j=1}^{n_i}||\boldsymbol{A}_{ij}\mathbf{x} - \boldsymbol{b}_{ij}||^2\right]$ with $\boldsymbol{A}_{ij} = i\boldsymbol{I}_d$, $\boldsymbol{\mu}_i \sim \mathcal{N}(0, \zeta^2(id)^{-2}\boldsymbol{I}_d)$, and $\boldsymbol{b}_{ij} \sim \mathcal{N}(\boldsymbol{\mu}_i, \sigma^2(id)^{-2}\boldsymbol{I}_d)$, where $\zeta^2$ controls the function similarity and $\sigma^2$ controls the stochastic noise (matching parameters in Corollary I). We depict the influence of shuffling on the convergence in Fig. 2.

In Fig. 2(a), we observe that in the high noise regime ($\sigma^2 = 1000$), shuffling gives a smaller reduction on the optimal error than when $\sigma^2 = 0$. In Fig. 2 (b), we tune the stepsize to reach to target accuracy $\varepsilon = 1.1 \cdot 10^{-6}$ with fewest

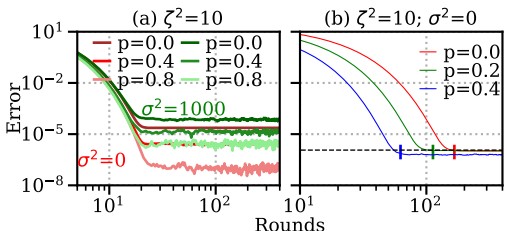

Figure 2: Convergence of $\frac{1}{n}\sum_{i=1}^{n}||\mathbf{x}_i^t - \mathbf{x}^\star||^2$. (a) With a fixed $\zeta^2$ and step size, shuffling reduces the optimal error more when the stochastic noise is low (b) When gradient dissimilarity $\zeta^2$ dominates the convergence, we obtain a super-linear speedup in the number of rounds to reach $\varepsilon$ by shuffling more data. The `vertical bar` shows the theoretical number of rounds to reach $\varepsilon$. The stepsize is tuned in (b).

rounds. Fig. 2 (b) shows that the empirical speedup matches the theoretical speedup as the observed and theoretical number of rounds (`vertical bars`) to reach $\varepsilon$ are very close—however, only when we replace $L$ in the theoretical bounds by $L_p$. This experiment shows that the practical performance of FedAvg can depend on $L_p$ (note in this experiment $\tilde{\mathcal{D}} = \mathcal{D}$), which hints that the pessimistic worst-case bounds from other works we have applied in Cor. I can potentially be improved.

## 3 SYNTHETIC DATA SHUFFLING

We have theoretically and empirically verified the benefit of adding shuffled data to each client. However, collecting data from clients goes against the goal of protecting user rights in FL. To still enjoy the benefit of shuffling and alleviate information leakage, we propose a practical framework that shuffles aggregated and more interpretable synthetic images from all the clients (see Fig. 1), shifting the problem into one of assuring privacy of the generator.

---

**Algorithm I** Fedssyn

1: **procedure** SYNTHETIC DATA GENERATION
2:     **for** client $i = 1, \ldots, N$ **in parallel do**
3:         sample $\rho \cdot n_i$ data points from $\mathcal{D}_i$
4:         train a *generator* $\mathcal{G}_i$
5:         generate $\tilde{\mathcal{D}}_i$ with $\tilde{n}$ samples
6:         Communicate $\tilde{\mathcal{D}}_i$ to the server
7:     **end for**
8:     Server *shuffles* the *aggregated* synthetic data $\{\tilde{\mathcal{D}}_1, \ldots, \tilde{\mathcal{D}}_N\}$ and *split* it to $N$ parts ($\{\hat{\mathcal{D}}_{si}\}$)
9:     Server *distributes* $\tilde{\mathcal{D}}_{si}$ ($|\tilde{\mathcal{D}}_{si}| = \tilde{n}$) to each client
10:     Run FL algorithm with the updated client dataset on each client $\hat{\mathcal{D}}_i := \mathcal{D}_i \cup \hat{\mathcal{D}}_{si}$
11: **end procedure**

**Algorithm II** Federated Averaging (FedAvg)

1: **procedure** FEDAVG
2:     **for** $r = 1, \ldots, R$ **do**
3:         Sample clients $S \subseteq \{1, \ldots, N\}$
4:         Send server model $\mathbf{x}$ to all clients $i \in S$
5:         **for** client $i \in S$ **in parallel do**
6:             initialise local model $\mathbf{y}_i \leftarrow \mathbf{x}$
7:             **for** $k = 1, \ldots, K$ **do**
8:                 $\mathbf{y}_i \leftarrow \mathbf{y}_i - \eta\nabla F_i(\mathbf{y}_i)$
9:             **end for**
10:         **end for**
11:         $\mathbf{x} \leftarrow \mathbf{x} + \frac{1}{|S|}\sum_{i \in S}(\mathbf{y}_i - \mathbf{x})$
12:     **end for**
13: **end procedure**

---

We denote the client dataset, client generated synthetic dataset, and shuffled synthetic dataset by $\mathcal{D}_i$, $\tilde{\mathcal{D}}_i$, and $\tilde{\mathcal{D}}$ (in accordance with the notation for the data distributions in Sec. 2). On each client $i$, we uniformly subsample $\rho \cdot n_i$ data points from the local dataset $\mathcal{D}_i$, given $\rho \in (0, 1]$ and $n_i = |\mathcal{D}_i|$, which we use to locally train a class-conditional generator. Using a subset rather than the entire dataset can alleviate the exposure of the class distribution from each client. Given the trained local generator, we generate $\tilde{\mathcal{D}}_i$ with $|\tilde{\mathcal{D}}_i| = \tilde{n}$ synthetic data, matching the class frequency in the subset. We assume that all the clients are willing to send their synthetic dataset to the server. The server

---

[1]In contrast to Koloskova et al. (2020) that consider the online setting, we generate here a finite set of samples on each client.

then shuffles the aggregated synthetic datasets $\cup_{i \in [N]} \tilde{\mathcal{D}}_i$ and distributes the data uniformly among the clients. In practice, when the number of synthetic data is large, their statistical dependency is negligible, and the synthetic data can be considered nearly iid. Reflecting this, we denote the *partitioned* shuffled synthetic dataset as $\tilde{\mathcal{D}}_{si}$ on each client, and the proportion of the synthetic dataset is calculated as $p := \frac{\tilde{n}}{n_i + \tilde{n}}$ (see Algorithm **I**).

We then run FL algorithms with the updated dataset on each client $\mathcal{D}_i \cup \tilde{\mathcal{D}}_{si}$. Our framework has no requirements for the FL algorithm, so we here briefly describe one of the most common FL algorithms: FedAvg (Konečný et al., 2016), as shown in Algorithm **II**. FedAvg mainly has two steps: local model updating and server aggregating. We initialize the server with a model **x**. In each communication round, each participating client $S \subseteq [N]$ receives a copy of the server parameter **x** and performs $K$ steps local updates (e.g., $K$ steps SGD). The updated model $\mathbf{y}_i$ is then sent to the server. FedAvg calculates the server model by averaging over the received client models, finishing one round of communication. We repeat this for $R$ rounds or until the target accuracy is achieved.

**Advantages of sending synthetic data rather than a generator for one-round:** 1) sending synthetic data can be more secure than sending the generator to the server since having access to generator's parameters gives the server more powerful attacking abilities (Wang et al., 2022b) 2) the clients can easily inspect the information shared with the server as the synthetic images are visually more interpretable than black-box generators 3) one-round synthetic data communication can be more secure than multi-round generator/synthetic data transmission (Xiong et al., 2023; Zhu et al., 2021) as the server cannot infer the local data distribution by identifying the difference between the updated synthetic data 4) depending on the generator, sending synthetic data can incur less communication cost, for example, transmitting the synthetic data leads to a 2×-30× reduction on the number of parameters communicated than transmitting the generator in our experiment.

## 4 EXPERIMENTAL SETUP

We show the effectiveness of our proposed method on CIFAR10 and CIFAR100 (Krizhevsky, 2009) image classification tasks. We partition the training dataset using Dirichlet distribution with a concentration parameter $\alpha$ to simulate the heterogeneous scenarios following Lin et al. (2020). Smaller $\alpha$ corresponds to higher data heterogeneity. We pick $\alpha \in \{0.01, 0.1\}$ as they are commonly used (Yu et al., 2022; Lin et al., 2020). Each client has a local dataset, kept fixed and local throughout the communication rounds. Experiments using MNIST and dSprites are in Appendix A.3.2 and A.4.

We use class-conditional DDPM Ho et al. (2020) on each client. We assume that all the clients participate in training DDPM by using 75% of their local data as the training data. Each client trains DDPM with a learning rate of 0.0001, 1000 diffusion time steps, 256 batch size, and 500 epochs. These hyperparameters are the same for all experiments. Once the training finishes, each client simulates $\tilde{n} = \frac{50000}{N}$ (there are 50K training images in CIFAR10 and CIFAR100) synthetic images with the label, which are then sent to the server. The server shuffles the aggregated synthetic data and then distributes it to each client equally.

We then perform federated optimization. We experiment with some of the popular FL algorithms including FedAvg (Konečný et al., 2016), FedProx (Sahu et al., 2018), SCAFFOLD (Karimireddy et al., 2020b), FedDyn (Acar et al., 2021), and FedPVR (Li et al., 2022a). We use VGG-11 for all the experiments. We train the FL algorithm with and without using the synthetic dataset to demonstrate the speedup obtained from using synthetic dataset quantitatively.

We use $N \in \{10, 40, 100\}$ as the number of clients and $C \in \{0.1, 0.2, 0.4, 1.0\}$ as the participation rate. For partial participation, we randomly sample $N \cdot C$ clients per communication round. We use a batch size of 256, 10 local epochs, and the number of gradient steps $\frac{10n_i}{256}$. We tune the learning rate from $\{0.01, 0.05, 0.1\}$ with the local validation dataset. The results are given as an average of three repeated experiments with different random initializations.

## 5 EXPERIMENTAL RESULTS

We show the performance of Fedssyn here. Our main findings are: 1) Sampling shuffled synthetic data into each client can significantly reduce the number of rounds used to reach a target accuracy

(1.6x-23x) and increase the Top-1 accuracy. 2) Fedssyn can reduce the number of communicated parameters to reach the target accuracy up to 95% than vanilla FedAvg. 3) Fedssyn is more robust across different levels of data heterogeneity compared to other synthetic-data based approaches 4) The quantified gradient dissimilarity and stochastic noise in DNN match the theoretical statements.

**Improved communication efficiency** We first report the required number of communication rounds to reach the target accuracy in Table 1. The length of the grey and red colour bar represents the required number of rounds to reach accuracy $m$ when the local dataset is $\mathcal{D}_i$ and $\mathcal{D}_i \cup \tilde{\mathcal{D}}_{si}$, respectively. The speedup is the ratio between the number of rounds in these two settings. We observe that adding shuffled synthetic data to each client can reduce the number of communication rounds to reach the target accuracy for all the cases.

Table 1: The required number of rounds to reach target accuracy $m$. The length of the grey and red bar equals to the number of rounds for experiments without the synthetic dataset ($\mathcal{D}_i$) and with the shuffled synthetic dataset ($\mathcal{D}_i \cup \tilde{\mathcal{D}}_{si}$), respectively. We annotate the length of the red bar, and the speedup (x) is the ratio between the number of rounds in these two settings. ">" means experiments with $\mathcal{D}_i$ as the local data cannot reach accuracy $m$ within 100 communication rounds. **Using shuffled synthetic dataset reduces the required number of communication round to reach the target accuracy in all cases.**

| | Full participation | | Partial participation | | | | | |
| | N=10 | | N=10 (C=40%) | | N=40 (C=20%) | | N=100 (C=10%) | |
| | $\alpha = 0.01$ | $\alpha = 0.1$ | $\alpha = 0.01$ | $\alpha = 0.1$ | $\alpha = 0.01$ | $\alpha = 0.1$ | $\alpha = 0.01$ | $\alpha = 0.1$ |
| | CIFAR10 | | | | | | | |
| | m = 44% | m = 66% | m = 44% | m = 66% | m = 44% | m = 66% | m = 44% | m = 66% |
| FedAvg | 4(22x) | 8(8.5x) | 4(> 25x) | 8(5.6x) | 12(8.3x) | 21(2.8x) | 19(4.9x) | 38(> 2.6x) |
| Scaffold | 4(9.3x) | 7(6.1x) | 5(> 20x) | 10(2.6x) | 12(6.8x) | 20(2.7x) | 31(> 3.2x) | 31(2.7x) |
| FedProx | 4(23.5x) | 8(5.3x) | 4(> 25x) | 9(8.2x) | 9(> 11.1x) | 15(4.7x) | 31(> 3.2x) | 37(> 2.7x) |
| FedDyn | 3(16.3x) | 5(14.8x) | 4(16.8x) | 7(10.7x) | 7(> 14.3x) | 12(> 5.3x) | 17(> 5.9x) | 31(> 3.2x) |
| FedPVR | 3(19.3x) | 7(4.3x) | 4(> 25x) | 8(9.5x) | 10(8.5x) | 21(2.8x) | 21(> 4.8x) | 35(> 2.9x) |
| | CIFAR100 | | | | | | | |
| | m = 30% | m = 40% | m = 20% | m = 20% | m = 30% | m = 40% | m = 30% | m = 30% |
| FedAvg | 15(5.9x) | 19(> 5.2x) | 17(> 5.9x) | 21(> 4.8x) | 37(> 2.7x) | 89(> 1.1x) | 70(> 1.4x) | 59(> 1.7x) |
| Scaffold | 14(2.9x) | 19(2.9x) | 15(3.6x) | 21(4.8x) | 24(> 4.2x) | 71(> 1.4x) | 61(1.6x) | 52(> 1.9x) |
| FedProx | 17(5.9x) | 23(4.3x) | 17(> 5.6x) | 23(> 4.3x) | 33(> 3.0x) | 100 + (−) | 75(> 1.3x) | 60(> 1.7x) |
| FedDyn | 10(3.6x) | 12(5.5x) | 13(> 7.7x) | 15(6.7x) | 25(> 4x) | 78(> 1.3x) | 61(> 1.6x) | 60(> 1.7x) |
| FedPVR | 14(6.1x) | 19(2.0x) | 15(> 6.6x) | 20(5.0x) | 23(> 4.3x) | 59(> 1.7x) | 71(> 1.4x) | 58(> 1.7x) |

**Reduced communication cost** Following Acar et al. (2021), we define the communication cost as the total number of parameters communicated between the clients and the server to reach a target accuracy. Assume the size of the synthetic data is $M_s$ and the size of the FL model is $M_c$, then the communication cost is calculated as $2M_s + 2RM_c$, where $R$ is the number of rounds to reach the target accuracy as shown in Table 1. In our experiment, $M_s$ is 2x-24x smaller than $M_c$, together with the drastically reduced number of communication rounds as shown in Table 1, we can save the communication cost up to 95%. See Table A.2 in the Appendix for the communication cost.

**Higher Top-1 accuracy** To verify that the accuracy improvement is mainly due to the changes in the convergence parameters rather than the increased number of images on each client, we compare experiments where the local dataset is $\mathcal{D}_i$, the mixture between local real and local synthetic data $\mathcal{D}_i \cup \tilde{\mathcal{D}}_i$, the mixture between local real and shuffled synthetic data $\mathcal{D}_i \cup \tilde{\mathcal{D}}_{si}$.

In Fig. 3, we observe that using the shuffled synthetic data can significantly improve the Top-1 accuracy in nearly all the cases compared to other baselines, indicates that the vast performance improvement is mainly due to the changes in the parameters in the convergence rate. We see that adding local synthetic data can sometimes lower the accuracy (e.g. `local+local synthetic` in CIFAR10 ($\alpha = 0.01$)). This is primarily because the local synthetic data amplifies the heterogeneity further and makes it more challenging for an FL algorithm to converge. Additionally, the large jump from using local synthetic data when the number of clients is large (e.g. CIFAR100 with 100 clients) is mainly because the increased number of data points per client can assist in local training. We observe a smaller performance improvement when the number of clients is large ($N = 100$) using Fedssyn, we can however further improve this by generating and shuffling more synthetic images.

**Sensitivity analysis** We here depict the sensitivity of FedAvg on the quality of the synthetic data using CIFAR10 in Fig. 4. Fig. 4 (a) shows that the number of images for training DDPM has a substantial impact on FL performance, which is reasonable, as we can acquire higher-quality

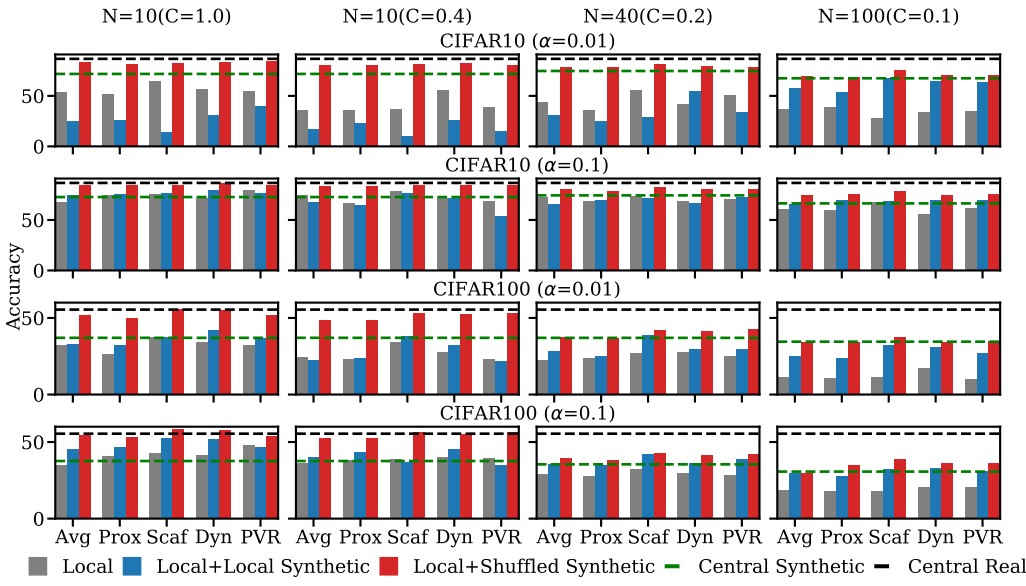

Figure 3: Top-1 accuracy. We compare the experiments where the local dataset is $\mathcal{D}_i$, $(1-p)\mathcal{D}_i + p\tilde{\mathcal{D}}_i$ (local+local synthetic data), and $(1-p)\mathcal{D}_i + p\hat{\mathcal{D}}_{si}$ (local+shuffled synthetic data). The black and green dotted lines represent the accuracy using the centralised real and synthetic data, respectively. Using shuffled synthetic data (red bar) boosts the Top-1 accuracy, and in some cases, even matches the centralised accuracy.

synthetic images if we use more images to train the generator. Fig. 4 (b) shows that when the number of clients is low, the synthetic images extracted from the early training checkpoints already have high quality. However, when the number of clients is high, the longer we train the generator, the better FL performance we can achieve. Note, in our experimental setup, a higher number of clients means $n_i = |\mathcal{D}_i|$ is smaller as $n_i = \frac{50,000}{N}$ where $N$ is the number of clients. Fig. 4 (c) shows that using DDPM is slightly better than GAN-based generators in terms of accuracy.

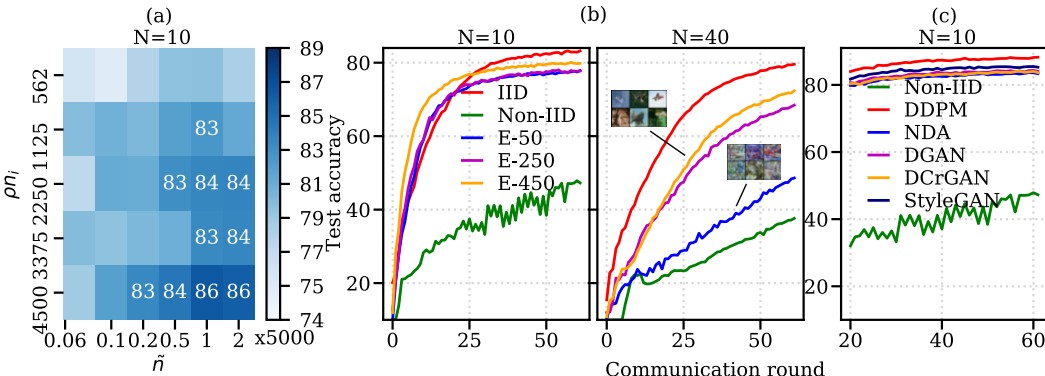

Figure 4: Sensitivity analysis using FedAvg and CIFAR10: (a) the influence of the number of images used for training the generator $\rho \cdot n_i$ and the number of synthetic images per client $\tilde{n}$ ($\alpha = 0.1$). We annotate the combination that performs better than the centralized baseline (b) example synthetic images for when E=50 and E=450 and the influence of the number of training epochs for the generator with 10 and 40 clients ($\alpha=0.01$). (c) the influence of using different generators ($\alpha = 0.01$)

**Comparison against other synthetic-data based approaches** We compare Fedssyn (FedAvg+shuffled synthetic data) with FedGEN (Zhu et al., 2021) and DENSE (Zhang et al., 2022) in Table 2. We train FedGEN and DENSE using the default hyperparameters as described in Zhu et al. (2021); Zhang et al. (2022). Following the setup in DENSE (Zhang et al., 2022), we use 10 clients with full participation, which is a commonly used FL setup in the literature (Li et al., 2022a;

2021; Yu et al., 2022). The low performance on DENSE is possibly due to the less carefully tuned random seeds. We can potentially improve the performance with more careful tuning. Comparably, Fedssyn achieves better Top-1 performance and is more robust against data heterogeneity levels. Compared to FedGEN which transmits the generator in every round, we only communicate the synthetic data *once*, which can be more secure and consume less cost. See Appendix A.2 for more information regarding the training details.

| | Fedssyn | FedGEN | DENSE |
|---|---|---|---|
| CIFAR10 ($\alpha$=0.01) | **83.0 (4)** | 47.6 (70) | 15.2 (-) |
| CIFAR10 ($\alpha$=0.1) | **84.1 (8)** | 78.6 (17) | 40.1 (-) |
| CIFAR100 ($\alpha$=0.01) | **52.0 (15)** | 35.1 (52) | 10.3 (-) |
| CIFAR100 ($\alpha$=0.1) | **54.4 (19)** | 41.0 (84) | 13.5 (-) |

Table 2: Top-1 accuracy (number of rounds to reach target accuracy) with 10 clients. We obtain better Top-1 accuracy and are faster to reach the target accuracy compared to other synthetic data-based methods.

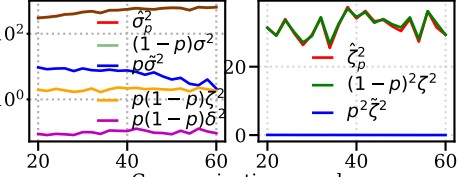

Figure 5: The empirical observation of stochastic noise and gradient dissimilarity matches the theoretical statement (experiment using CIFAR10, 10 clients, $\alpha = 0.1$, p=0.06)

**Parameters in the convergence rate** We investigate the impact of using shuffled synthetic data on the parameters in the convergence rate for DNN-based FedAvg in Fig. 5. We use CIFAR10, 10 clients with full participation, $\alpha = 0.1$. We observe that $(1 - p)\sigma^2$ dominates over other terms in the effective stochastic noise, which means the first term in the convergence rate for non-convex function in Corollary I can be simplified as $\mathcal{O}\left((1-p)\sigma^2 L_p/(n\varepsilon^2)\right)$ in this experimental setup. For $\hat{\zeta}_p^2$, the empirical result also matches the theory. These results show that in this experimental setup, adding shuffled synthetic data reduces both stochastic noise and function dissimilarity, and lead to a greater accuracy improvement and vast reduction on the number of rounds to reach the target accuracy. See Appendix A.3.5 for more information.

**Practical implications** Our proposed framework Fedssyn is more suitable for applications such as disease diagnosis and explosives detection that usually require high precision seeing that Fedssyn can even reach the centralized accuracy in some of the high data heterogeneity scenarios and the saved communication cost is enormous (up to 95%). Suppose the client is resource-constraint, a good strategy is to start with generating fewer synthetic images since shuffling even a small percentage (e.g. p=0.06) of the synthetic images into each client can already improve the Top-1 accuracy by 14%-20% (Fig. 4) and a larger $p$ usually corresponds to a better accuracy until it saturates, e.g. reach the iid accuracy. Additionally, the client has the option of checking the synthetic images and only sharing the less sensitive synthetic images to the server to alleviate the information leakage.

**Limitations** Training the generator locally may pose a computation requirement for each device. We can mitigate this issue with methods such as quantization (Alistarh et al., 2016) and compression (Li et al., 2020; Mishchenko et al., 2019). Sending the synthetic images may also leak the training data distribution. One possible solution to improve the utility of Fedssyn is to make the generator and federated learning algorithms differential private Yoon et al. (2021; 2019); Abadi et al. (2016). As this introduces complicated trade-off between model utility, synthetic image quality, and privacy budget and can complicate the theoretical understanding, we leave this as future work.

## 6 CONCLUSION

In this paper, we have rigorously analyzed the relation between data heterogeneity and the parameters in the convergence rate in FedAvg under standard stochastic noise and gradient dissimilarity assumptions. While previous works usually qualitatively represent data heterogeneity with gradient dissimilarity, we proposed a more quantifiable and precise characterization of how the changes in the data heterogeneity impact the parameters in the convergence rate using shuffling.

Our work paves the way for a better understanding of the seemingly unreasonable effect of data heterogeneity in FL. The theoretical findings inspired us to present a more practical framework, which shuffles locally generated synthetic data, achieving nearly as good performance as centralized

learning, even in some cases when the data heterogeneity is high. Our work reveals the importance of data distribution matching in FL, opening up future research directions.

## REPRODUCIBILITY STATEMENT

We provide downloadable source code as part of the supplementary material. This code allows to reproduce our experimental evaluations show in the main part of the manuscript. The code for the additional verification on the dSprites dataset in Appendix A.4 is not part of the submitted file. We will make this code and the genenerated datasets (such as in Figure A.7) available on a public github repository.

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

## A APPENDIX

We first provide the proof for the lemma. We then show the comparison with other synthetic-data based works. Following that, we show the extra experimental details and experimental results on two other dataset MNIST and dSprites.

### A.1 PROOF

We study the distributed stochastic optimization problem:

$$f(\mathbf{x}, \mathcal{D}) := \mathbb{E}_{\xi \sim \mathcal{D}}[F(\mathbf{x}, \xi)], \quad f(\mathbf{x}, \mathcal{D}_i) := \mathbb{E}_{\xi \sim \mathcal{D}_i}[F(\mathbf{x}, \xi)]$$

where $\xi$ can be a random data point or a random function, $\mathcal{D}_i$ denotes the distribution of $\xi$ on client $i$ and $\mathcal{D}$ denotes the distribution of $\xi$ over all the clients.

We argue that it is possibly not be feasible to assume access to the distribution $\mathcal{D}$ as collecting data from clients goes against the goal of protecting privacy in federated learning. Instead, we use an approximation (e.g. synthetic data) $\tilde{\mathcal{D}}$ for carrying out the proof. We will demonstrate the impact of the difference between $\mathcal{D}$ and $\tilde{\mathcal{D}}$ in the following sections.

Given the effective data $\mathcal{D}_i \cup \tilde{\mathcal{D}}$ on each client we can formulate the optimization problem as:

$$F(\mathbf{x}, \xi \sim \mathcal{D}_i \cup \tilde{\mathcal{D}}; b) := \begin{cases} F(\mathbf{x}, \xi \sim \mathcal{D}_i \mid b = 0) & \text{with probability } 1 - p \\ F(\mathbf{x}, \xi \sim \tilde{\mathcal{D}} \mid b = 1) & \text{with probability } p \end{cases} \tag{A.1}$$

where $b$ is a random variable that indicates where a data point is drawn from. When $b = 0$ (with probability $1 - p$), we assume that we draw from a worker's own data $\mathcal{D}_i$. When $b = 1$ (with probability $p$), we assume that we draw from the uniform distribution $\tilde{\mathcal{D}}$. Take the expectation with respect to $b$, we have:

$$F(\mathbf{x}, \xi \sim \mathcal{D}_i \cup \tilde{\mathcal{D}}) = \mathbb{E}_b[F(\mathbf{x}, \xi; b)] = (1 - p)F(\mathbf{x}, \xi \sim \mathcal{D}_i) + pF(\mathbf{x}, \xi \sim \tilde{\mathcal{D}}) \tag{A.2a}$$

$$\begin{aligned} f(\mathbf{x}, \mathcal{D}_i \cup \tilde{\mathcal{D}}) = \mathbb{E}_{\xi \sim \mathcal{D}_i \cup \tilde{\mathcal{D}}}[F(\mathbf{x}, \xi)] &= (1 - p)\mathbb{E}_{\xi \sim \mathcal{D}_i}[F(\mathbf{x}, \xi)] + p\mathbb{E}_{\xi \sim \tilde{\mathcal{D}}}[F(\mathbf{x}, \xi)] \\ &= (1 - p)f(\mathbf{x}, \mathcal{D}_i) + pf(\mathbf{x}, \tilde{\mathcal{D}}) \end{aligned} \tag{A.2b}$$

$$\begin{aligned} f(\mathbf{x}, \mathcal{D} \cup \tilde{\mathcal{D}}) = \mathbb{E}_{\xi \sim \mathcal{D} \cup \tilde{\mathcal{D}}}[F(\mathbf{x}, \xi)] &= (1 - p)\mathbb{E}_{\xi \sim \mathcal{D}}[F(\mathbf{x}, \xi)] + p\mathbb{E}_{\xi \sim \tilde{\mathcal{D}}}[F(\mathbf{x}, \xi)] \\ &= (1 - p)f(\mathbf{x}, \mathcal{D}) + pf(\mathbf{x}, \tilde{\mathcal{D}}) \end{aligned} \tag{A.2c}$$

**Lemma A-1** (variance with probability). *If the random variable $X$ is discrete with each element $x_i$ appearing with a probability $p_i$ such that $p(x_i) = p_i$, then:*

$$\mathbb{E}||X - \mu||^2 = \sum_{i=1}^{n} p_i ||x_i - \mu||^2, \quad \mu := \frac{1}{n} \sum_{i=1}^{n} p_i x_i.$$

**Lemma A-2** (shuffled gradient dissimilarity). *If Assumption 1 holds, then $\forall \mathbf{x} \in \mathbb{R}^d$:*

$$\hat{\zeta}_p^2 := \mathbb{E}||\nabla f(\mathbf{x}, \mathcal{D}_i \cup \tilde{\mathcal{D}}) - \nabla f(\mathbf{x}, \mathcal{D} \cup \tilde{\mathcal{D}})||^2 \leq (1 - p)^2 \zeta^2,$$

*where here the expectation is taken over a random index $i \in [N]$.*

*Proof.* Given the definition from Eq. A.2, we have:

$$\begin{aligned} \hat{\zeta}_p^2 : &= \mathbb{E}||\nabla f(\mathbf{x}, \mathcal{D}_i \cup \tilde{\mathcal{D}}) - \nabla f(\mathbf{x}, \mathcal{D} \cup \tilde{\mathcal{D}})||^2 \\ &= \mathbb{E}||(1 - p)\nabla f(\mathbf{x}, \mathcal{D}_i) + p\nabla f(\mathbf{x}, \tilde{\mathcal{D}}) - (1 - p)\nabla f(\mathbf{x}, \mathcal{D}) - p\nabla f(\mathbf{x}, \tilde{\mathcal{D}})||^2 \\ &= (1 - p)^2 \mathbb{E}||\nabla f(\mathbf{x}, \mathcal{D}_i) - \nabla f(\mathbf{x}, \mathcal{D})||^2 \\ &\leq (1 - p)^2 \zeta^2. \end{aligned} \tag{A.3}$$

$\square$

**Lemma A-3** (shuffled stochastic noise). *If Assumption 1, 2, and 4 hold and let $\tilde{\sigma}^2_{avg}$ being the stochastic noise from using $\tilde{\mathcal{D}}$ such that $\mathbb{E}_{\xi \sim \tilde{\mathcal{D}}}||\nabla F(\mathbf{x}, \xi) - f(\mathbf{x}, \tilde{\mathcal{D}})||^2 \leq \tilde{\sigma}^2_{avg}$, then $\forall \mathbf{x} \in \mathbb{R}^d$, we have:*

$$\hat{\sigma}^2_p := \mathbb{E}_{\xi \sim \mathcal{D}_i \cup \tilde{\mathcal{D}}}||\nabla F(\mathbf{x}, \xi) - \nabla f(\mathbf{x}, \mathcal{D}_i \cup \tilde{\mathcal{D}})||^2 \leq (1-p)\sigma^2 + p\tilde{\sigma}^2_{avg} + p(1-p)||\nabla f(\mathbf{x}, \mathcal{D}_i) - \nabla f(\mathbf{x}, \tilde{\mathcal{D}})||^2$$

$$\mathbb{E}[\hat{\sigma}^2_p] \leq (1-p)\sigma^2 + p\tilde{\sigma}^2_{avg} + p(1-p)\zeta^2 + p(1-p)\delta^2$$

*Proof.* Based on the definition in Eq. A.2, we have:

$$\hat{\sigma}^2_p := \mathbb{E}_{\xi \sim \mathcal{D}_i \cup \tilde{\mathcal{D}}}||\nabla F(\mathbf{x}, \xi) - \nabla f(\mathbf{x}, \mathcal{D}_i \cup \tilde{\mathcal{D}})||^2$$

$$= (1-p)\underbrace{\mathbb{E}_{\xi \sim \mathcal{D}_i}||\nabla F(\mathbf{x}, \xi) - \nabla f(\mathbf{x}, \mathcal{D}_i \cup \tilde{\mathcal{D}})||^2}_{\mathcal{A}_1} + p\underbrace{\mathbb{E}_{\xi \sim \tilde{\mathcal{D}}}||\nabla F(\mathbf{x}, \xi) - \nabla f(\mathbf{x}, \mathcal{D}_i \cup \tilde{\mathcal{D}})||^2}_{\mathcal{A}_2}$$

We use Lemma A-1 in the last equality. We next give the bound for $\mathcal{A}_1$ and $\mathcal{A}_2$.

$$\mathcal{A}_1 := \mathbb{E}_{\xi \sim \mathcal{D}_i}||\nabla F(\mathbf{x}, \xi) - \nabla f(\mathbf{x}, \mathcal{D}_i) + \nabla f(\mathbf{x}, \mathcal{D}_i) - \nabla f(\mathbf{x}, \mathcal{D}_i \cup \tilde{\mathcal{D}})||^2$$

$$= \mathbb{E}_{\xi \sim \mathcal{D}_i}||\nabla F(\mathbf{x}, \xi) - \nabla f(\mathbf{x}, \mathcal{D}_i)||^2 + ||\nabla f(\mathbf{x}, \mathcal{D}_i) - \nabla f(\mathbf{x}, \mathcal{D}_i \cup \tilde{\mathcal{D}})||^2$$

$$\leq \sigma^2 + ||\nabla f(\mathbf{x}, \mathcal{D}_i) - (1-p)f(\mathbf{x}, \mathcal{D}_i) - pf(\mathbf{x}, \tilde{\mathcal{D}})||^2$$

$$\leq \sigma^2 + p^2||\nabla f(\mathbf{x}, \mathcal{D}_i) - \nabla f(\mathbf{x}, \tilde{\mathcal{D}})||^2$$

Similarly, we can bound $\mathcal{A}_2$ as:

$$\mathcal{A}_2 := \mathbb{E}_{\xi \sim \tilde{\mathcal{D}}}||\nabla F(\mathbf{x}, \xi) - \nabla f(\mathbf{x}, \mathcal{D}_i \cup \tilde{\mathcal{D}})||^2$$

$$= \mathbb{E}_{\xi \sim \tilde{\mathcal{D}}}||\nabla F(\mathbf{x}, \xi) - \nabla f(\mathbf{x}, \tilde{\mathcal{D}})||^2 + ||\nabla f(\mathbf{x}, \tilde{\mathcal{D}}) - \nabla f(\mathbf{x}, \mathcal{D}_i \cup \tilde{\mathcal{D}})||^2$$

$$\leq \tilde{\sigma}^2_{avg} + ||\nabla f(\mathbf{x}, \tilde{\mathcal{D}}) - (1-p)f(\mathbf{x}, \mathcal{D}_i) - pf(\mathbf{x}, \tilde{\mathcal{D}})||^2$$

$$= \tilde{\sigma}^2_{avg} + (1-p)^2||\nabla f(\mathbf{x}, \mathcal{D}_i) - \nabla f(\mathbf{x}, \tilde{\mathcal{D}})||^2$$

Taking the bound for $\mathcal{A}_1$ and $\mathcal{A}_2$ back to the expression of $\hat{\sigma}^2_p$, we have:

$$\hat{\sigma}^2_p \leq (1-p)\sigma^2 + p\tilde{\sigma}^2_{avg} + p(1-p)||\nabla f(\mathbf{x}, \mathcal{D}_i) - \nabla f(\mathbf{x}, \tilde{\mathcal{D}})||^2$$

If we take the expectation w.r.t the clients for both sides, we then have:

$$\mathbb{E}[\hat{\sigma}^2_p] \leq (1-p)\sigma^2 + p\tilde{\sigma}^2_{avg} + p(1-p)\mathbb{E}||\nabla f(\mathbf{x}, \mathcal{D}_i) - \nabla f(\mathbf{x}, \tilde{\mathcal{D}})||^2$$

$$= (1-p)\sigma^2 + p\tilde{\sigma}^2_{avg} + p(1-p)\mathbb{E}||\nabla f(\mathbf{x}, \mathcal{D}_i) - \nabla f(\mathbf{x}, \mathcal{D}) + \nabla f(\mathbf{x}, \mathcal{D}) - \nabla f(\mathbf{x}, \tilde{\mathcal{D}})||^2$$

$$= (1-p)\sigma^2 + p\tilde{\sigma}^2_{avg} + p(1-p)\mathbb{E}||\nabla f(\mathbf{x}, \mathcal{D}_i) - \nabla f(\mathbf{x}, \mathcal{D})||^2 + p(1-p)||\nabla f(\mathbf{x}, \mathcal{D}) - \nabla f(\mathbf{x}, \tilde{\mathcal{D}})||^2$$

$$\leq (1-p)\sigma^2 + p\tilde{\sigma}^2_{avg} + p(1-p)\zeta^2 + p(1-p)||\nabla f(\mathbf{x}, \mathcal{D}) - \nabla f(\mathbf{x}, \tilde{\mathcal{D}})||^2$$

$$\leq (1-p)\sigma^2 + p\tilde{\sigma}^2_{avg} + p(1-p)\zeta^2 + p(1-p)\delta^2$$

$$\text{(A.4)}$$

The inequalities use the Assumption 1 and 4. If $\tilde{\mathcal{D}} = \mathcal{D}$, then $\delta^2 = 0$ by definition. We can then rewrite the bound for the stochastic noise as $\mathbb{E}[\hat{\sigma}^2_p] \leq (1-p)\sigma^2 + p\sigma^2_{avg} + p(1-p)\zeta^2$ □

**Lemma A-4** (shuffled smoothness). *Let $L_{\max}$ and $L_{avg}$ denote the maximum and average smoothness, i.e. it holds*

$$||\nabla f(\mathbf{x}, \mathcal{D}_i) - \nabla f(\mathbf{y}, \mathcal{D}_i)|| \leq L_{\max} ||\mathbf{x} - \mathbf{y}||, \qquad \forall \mathbf{x}, \mathbf{y} \in \mathbb{R}^d, i \in [n],$$

*and*

$$||\nabla f(\mathbf{x}, \mathcal{D}) - \nabla f(\mathbf{y}, \mathcal{D})|| \leq L_{avg} ||\mathbf{x} - \mathbf{y}||, \qquad \forall \mathbf{x}, \mathbf{y} \in \mathbb{R}^d.$$

*and*

$$\left|\left|\nabla f(\mathbf{x}, \tilde{\mathcal{D}}) - \nabla f(\mathbf{y}, \tilde{\mathcal{D}})\right|\right| \leq \tilde{L}_{avg} ||\mathbf{x} - \mathbf{y}||, \qquad \forall \mathbf{x}, \mathbf{y} \in \mathbb{R}^d$$

*Then we have:*

$$\mathbb{E}L_p \leq (1-p)L_{\max} + p\tilde{L}_{avg},$$

*where $\mathbb{E}L_p$ denotes an upper bound on the smoothness constant of $f(\mathbf{x}, \mathcal{D}_i \cup \tilde{\mathcal{D}})$.*

*Proof.* Given the client index $i$, by definition, we have $f(\mathbf{x}, \mathcal{D}_i \cup \tilde{\mathcal{D}}) = (1-p)f(\mathbf{x}, \mathcal{D}_i) + pf(\mathbf{x}; \tilde{\mathcal{D}})$,

$$
\begin{aligned}
||\nabla f(\mathbf{x}, \mathcal{D}_i \cup \tilde{\mathcal{D}}) - \nabla f(\mathbf{y}, \mathcal{D}_i \cup \tilde{\mathcal{D}})|| &= ||(1-p)f(\mathbf{x}, \mathcal{D}_i) + pf(\mathbf{x}; \tilde{\mathcal{D}}) - \nabla f(\mathbf{y}, \mathcal{D}_i \cup \tilde{\mathcal{D}})|| \\
&= ||(1-p)\left(\nabla f(\mathbf{x}, \mathcal{D}_i) - \nabla f(\mathbf{y}, \mathcal{D}_i)\right) + p\left(\nabla f(\mathbf{x}; \tilde{\mathcal{D}}) - \nabla f(\mathbf{y}; \tilde{\mathcal{D}})\right)|| \\
&\leq (1-p)||\nabla f(\mathbf{x}, \mathcal{D}_i) - \nabla f(\mathbf{y}, \mathcal{D}_i)|| + p||\nabla f(\mathbf{x}; \tilde{\mathcal{D}}) - \nabla f(\mathbf{y}; \tilde{\mathcal{D}})|| \\
&\leq \left((1-p)L_{\max} + pL_{\text{avg}}\right)||\mathbf{x} - \mathbf{y}||.
\end{aligned}
$$

$\square$

## A.2 EXTRA RELATED WORK COMPARISON

Table A.1: Comparison with the existing approaches that use synthetic data in FL. Our proposed method is easy to use and has removed many complicated components from the existing methods.

| | Local $\mathcal{G}_i$ | Server $\mathcal{G}$ | $\tilde{\mathcal{D}}(r)$ | $\mathbf{x} \leftarrow \tilde{\mathcal{D}}$ | Pseudo-label |
|---|:---:|:---:|:---:|:---:|:---:|
| FedDM Xiong et al. (2023) | ✓ | ✗ | ✓ | ✓ | ✗ |
| FedGEN Zhu et al. (2021) | ✗ | ✓ | ✓ | ✗ | ✗ |
| Dense Zhang et al. (2022) | ✗ | ✓ | ✗ | ✓ | ✗ |
| SDA-FL Li et al. (2022b) | ✓ | ✗ | ✓ | ✓ | ✓ |
| ours | ✓ | ✗ | ✗ | ✗ | ✗ |

[1] $\tilde{\mathcal{D}}(r)$: transmit the synthetic dataset every round

[2] $\mathbf{x} \leftarrow \tilde{\mathcal{D}}$: updates the server model $\mathbf{x}$ with $\tilde{\mathcal{D}}$ every round

Local $\mathcal{G}_i$ and Server $\mathcal{G}$ means local and server generators for generating synthetic datasets. $\tilde{\mathcal{D}}(r)$ means the synthetic data is transmitted between the client and server at every communication round, which can incur extra communication costs. Updating the server model with the synthetic data at every communication round $\mathbf{x} \leftarrow \tilde{\mathcal{D}}$ requires the synthetic data to be of high quality Lin et al. (2020); Sehwag et al. (2022), as updating the server model with low-quality synthetic data or synthetic data with a different distribution than the actual local data may discard the valuable information it has learned.

**Experimental details for replicating the FedGEN and DENSE results** For FedGEN, we follow the default hyperparameter setup as described in the paper Zhu et al. (2021) and the official repository. We tune the optimal stepsize from $\{0.05, 0.1, 0.2\}$. We run FedGEN for 100 communication rounds and report the final Top-1 accuracy using the server model on the test dataset. For DENSE, we follow the hyperparameter setup as described in the paper Zhang et al. (2022) and the official GitHub repository. We have tried our best to improve the quality of the server model obtained after one round of communication as have done in Zhang et al. (2022). However, we still observe that such server model is not sufficiently good for the global distillation learning, especially when the data heterogeneity is high ($\alpha = 0.01$). Therefore, we performed ten rounds of communication and then used the server model for global distillation learning. We can possibly improve the performance of FedGEN and DENSE with more careful tuning of the random seeds and other hyperparameters.

## A.3 EXTRA EXPERIMENTAL DETAILS

### A.3.1 SIMULATED IMAGES

### A.3.2 EXPERIMENTAL RESULTS ON MNIST DATASET

We consider a distributed multi-class classification problem on MNIST dataset given $(A_i, \boldsymbol{b}_i)$ as the dataset on worker $i$ with:

$$
\mathbf{y}_i = \text{softmax}(A_i\mathbf{x}), \quad A_i \sim \mathbb{R}^{n_i \times 784}, \mathbf{x} \in \mathbb{R}^{784 \times 10}, \mathbf{y}_i \in \mathbb{R}^{n_i \times 10} \tag{A.5a}
$$

$$
f_i(\mathbf{x}) := \frac{1}{n_i} \sum_{j=1}^{n_i} \frac{1}{10} \sum_{k=1}^{10} (\boldsymbol{b}_{ijk} \log(\mathbf{y}_{ijk})) \tag{A.5b}
$$

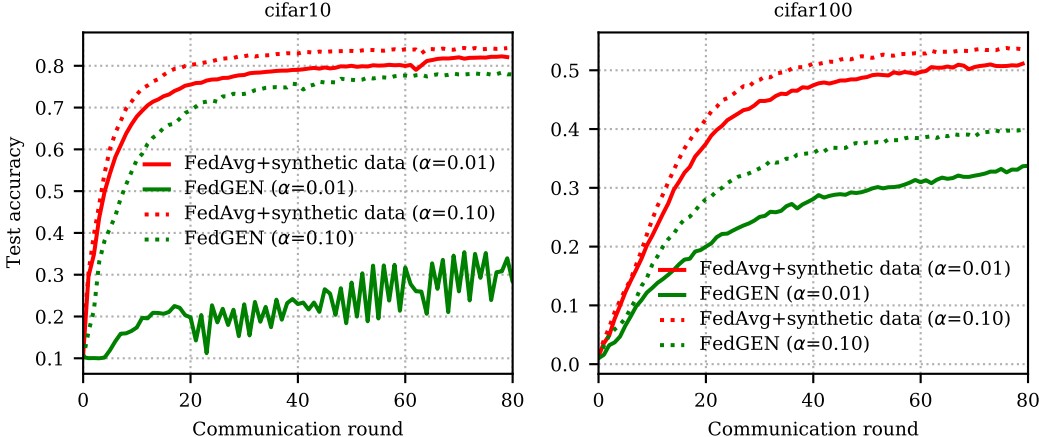

Figure A.1: Test accuracy for two methods FedAvg + synthetic data and FedGEN. Comparably, our proposed method reaches to a better test accuracy faster than FedGEN. The unstable performance of when $\alpha = 0.01$ on CIFAR10 dataset using FedGEN can be potentially improved with more careful hyperparameter tuning.

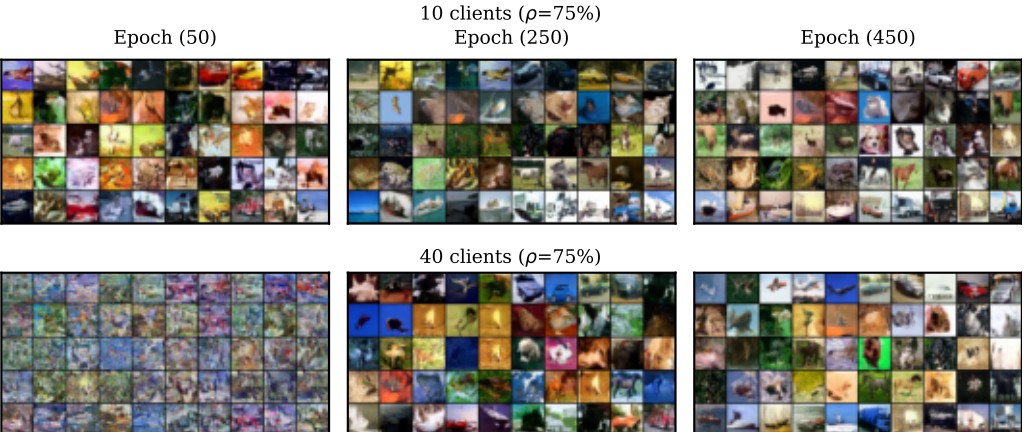

Figure A.2: Example simulated images from different training epochs on CIFAR10. The number of training images per client is 3375 for the top row and 875 for the bottom row. When the number of clients is small and the number of training images for training DDPM is high, the images extracted from Epoch 50 are already high quality. However, when the number of clients is high, the longer we train the DDPM, the better images we can obtain.

We construct the training dataset by subsampling 1024 images from each class from the MNIST training dataset. We use the test dataset to evaluate the performance of the server model (10000 images). We consider two scenarios for splitting the dataset across workers: 1) `iid`, where each worker has a similar number of images across classes 2) `split by class`, where each worker can only see images from a single class. We set 10 workers, and each worker has 1024 images $n_i = 1024$. We experiment with a different number of local epochs $E$ on each worker and shuffling percentage $p$. For a given $p$, we subsample $n_i p$ data points randomly from each worker and allocate them in $\tilde{\mathcal{D}}$. We then shuffle $\tilde{\mathcal{D}}$ and distribute it equally to all workers such that each worker has $n_i p$ shuffled data points and $n_i(1-p)$ local data points. We choose to control the stochastic noise $\bar{\sigma}^2$ by adding Gaussian noise to every gradient:

$$\mathbf{y}_i \leftarrow \mathbf{y}_i - \eta(\nabla f_i(\mathbf{y}_i) + s), \quad s \sim \mathcal{N}(0, \frac{\bar{\sigma}^2}{784}) \tag{A.6}$$

Fig. A.3 shows the results. When the dataset is IID across workers, adding the shuffled dataset to each worker does not influence the learning speed. This agrees with the lemmas that we have shown in the main text. When the dataset is non-iid across workers, shuffling a small percentage of the

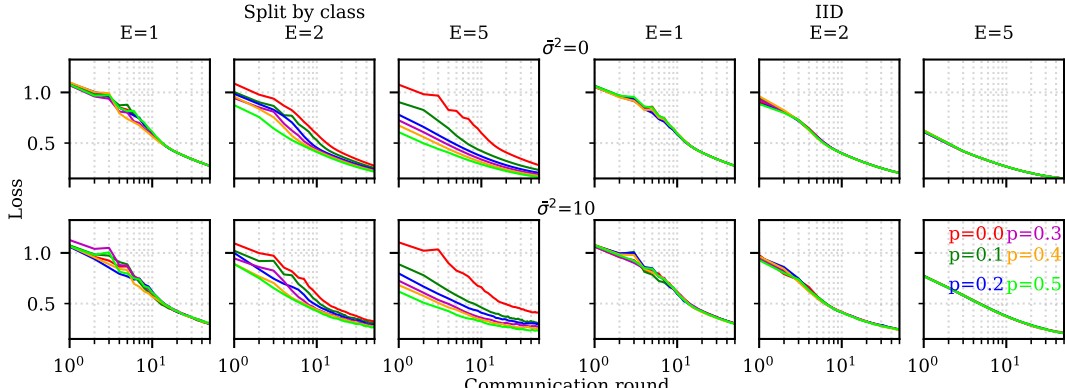

Figure A.3: Test loss using the server model over communication rounds. When the dataset is IID across workers, adding shuffled dataset does not influence the learning speed. However, when workers have heterogeneous datasets (`split by class`), shuffling a small percentage of the dataset can improve the convergence speed, especially when the number of local epochs is high (e.g., $E = 5$)

local dataset can highly improve the convergence, especially when the number of local epochs is high, e.g., $E = 5$. We can achieve a slightly better speedup when the stochastic noise is non-zero, which is most of the cases in the neural network training where we use mini-batch SGD.

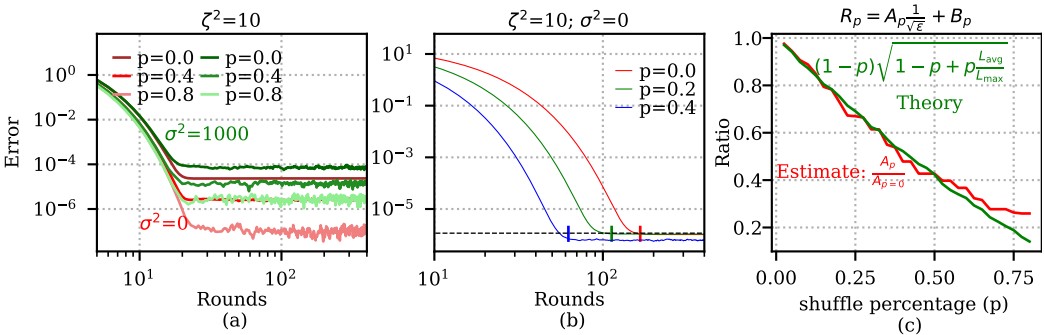

Figure A.4: Convergence of $\frac{1}{n} \sum_{i=1}^{n} ||\mathbf{x}_i^t - \mathbf{x}^\star||^2$. (a) With a fixed $\zeta^2$ and step size, shuffling reduces the optimal error more when the stochastic noise is low (b) When gradient dissimilarity $\zeta^2$ dominates the convergence, we obtain a super-linear speedup in the number of rounds to reach $\varepsilon$ by shuffling more data. The vertical bar shows the theoretical number of rounds to reach $\varepsilon$ (c) The estimated ratio matches the theoretical ratio when the shuffle percentage is small (e.g., $p \leq 0.6$). The term $\sqrt{1 - p + p\frac{L_{\text{avg}}}{L_{\text{max}}}}$ comes from the Lipschitz constant ratio $\sqrt{\frac{L_p}{L_{p=0}}}$.

### A.3.3  EXTRA EXPLANATION FOR FIG.2

We consider a distributed least squares objective $f(\mathbf{x}) := \frac{1}{n} \sum_{i=1}^{n} \left[ f_i(\mathbf{x}) := \frac{1}{2n_i} \sum_{j=1}^{n_i} ||\boldsymbol{A}_{ij}\mathbf{x} - \boldsymbol{b}_{ij}||^2 \right]$ with $\boldsymbol{A}_{ij} = i\boldsymbol{I}_d$, $\boldsymbol{\mu}_i \sim \mathcal{N}(0, \zeta^2(id)^{-2}\boldsymbol{I}_d)$, and $\boldsymbol{b}_{ij} \sim \mathcal{N}(\boldsymbol{\mu}_i, \sigma^2(id)^{-2}\boldsymbol{I}_d)$, where $\zeta^2$ controls the function similarity and $\sigma^2$ controls the stochastic noise (matching parameters in Corollary I). On each worker, we generate $n_i = 100$ pairs of $\{\boldsymbol{A}_{ij}, \boldsymbol{b}_{ij}\}$ with $d = 25$. We randomly sample $p \cdot n_i$ pairs of $\{\boldsymbol{A}_{ij}, \boldsymbol{b}_{ij}\}$ from all the clients. We then aggregate, shuffle, and redistribute them equally to each client such that the number of functions per client is still $n_i$. We depict the influence of shuffling on different parameters in Fig. A.4. To concisely match out theory, we need to consider that also the curvature of the function changes and measure $L_p$, $L_{\text{max}} := \max_i \left\| \frac{1}{n_i} \sum_{j=1}^{n_i} \boldsymbol{A}_{ij}^T \boldsymbol{A}_{ij} \right\|$ and $L_{\text{avg}} := \frac{1}{n} \sum_{i=1}^{n} \left\| \frac{1}{n_i} \sum_{j=1}^{n_i} \boldsymbol{A}_{ij}^T \boldsymbol{A}_{ij} \right\|$.

Fig. A.4 (a) shows the influence of shuffling when we vary the stochastic noise given a fixed $\zeta^2$ and stepsize. We observe that in the high-noise regime, shuffling gives a smaller reduction on the

optimal error than when $\sigma^2 = 0$ as $\mathcal{O}\left(\frac{(1-p)\sigma^2 + p\sigma_{\text{avg}}^2}{\varepsilon}\right)$ tends to dominate no matter how much

data we shuffle. Fig. A.4 (b) shows the influence of shuffling when $\sigma^2 = 0$ and $\mathcal{O}\left(\frac{\sqrt{L_p}\tau(1-p)\zeta}{\sqrt{\varepsilon}}\right)$

dominates the convergence. We tune the step size for each experiment to reach the target accuracy ($\varepsilon = 1.1 \cdot 10^{-6}$) with the fewest rounds. The theoretical number of rounds required to reach $\varepsilon$,

indicated by the vertical bars, were determined by calculating $R_p = \frac{A_p}{\sqrt{\varepsilon}} + B_p$, where $A_p := \frac{\sqrt{L_p}\tau\hat{\zeta}_p}{\mu}$

is the coefficient that depends on $\zeta^2$ and $L_p$. To estimate $A_p$, we measure the number of rounds to reach different accuracies and fit a linear line between $R_p$ and $\frac{1}{\sqrt{\varepsilon}}$. Fig. A.4 (b) shows that the empirical speedup matches the theoretical speedup as the observed and theoretical number of rounds to reach the target accuracy are very close. For better visualization, we compare the theoretical and estimated ratio in Fig. A.4 (c) given $\sigma^2 = 0$. Fig. A.4(c) shows the ratio between the coefficient $\frac{A_p}{A_{p=0}}$

which boils down to $\mathcal{O}\left(\frac{\sqrt{L_p}\zeta_p}{\sqrt{L_{p=0}}\zeta_{p=0}}\right)$. The nearly matching lines in Fig. A.4 (c), especially when $p$

is small, verify our theoretical statement about the impact of data heterogeneity on the convergence parameters. When $p$ is large, $\sigma^2$ and the constant term tend to dominate the convergence. A linear line between $R_p$ and $\frac{1}{\sqrt{\varepsilon}}$ is incomplete to explain the convergence rate.

### A.3.4 COMMUNICATION COST

We calculate the number of transmitted parameters when FedAvg is used in Table A.2, which shows that we can reduce the number of transmitted parameters up to 95% to achieve a certain level of accuracy. For methods such as SCAFFOLD Karimireddy et al. (2020b) and FedPVR Li et al. (2022a), we also need to take into account the transmitted control variates when we calculate the number of transmitted parameters per round.

Table A.2: The saved communication cost of using Fedssyn compared to the vanilla FedAvg (percentage between the number of transmitted parameters). $M_s$ is the size of the transmitted synthetic data and $M_c$ is the size of the FL model. $M_c$ in our experiment is around 37.2 MB. Fedssyn transmits significantly less number of parameters compared to the vanilla FedAvg to reach the same level of accuracy.

| Number of client (participation rate) | CIFAR10 | | CIFAR100 | |
|---|---|---|---|---|
| | $\alpha = 0.01$ | $\alpha = 0.1$ | $\alpha = 0.01$ | $\alpha = 0.1$ |
| N=10 (C=1.0) ($M_s = 15.5$MB) | 95.0% | 87.6% | 82.5% | >80.2% |
| N=10 (C=0.4) ($M_s = 15.5$MB) | >95.6% | 80.9% | >82.6% | >78.6% |
| N=40 (C=0.2) ($M_s = 0.39$MB) | 87.9% | 63.8% | >62.6% | >8.2% |
| N=100 (C=0.1) ($M_s = 0.16$MB) | 79.6% | >61.2% | >28.6% | >41.0% |

### A.3.5 CONVERGENCE PARAMETERS IN DNN EXPERIMENTS

We here investigate the impact of using shuffled synthetic data on the parameters in the convergence rate for DNN-based FedAvg in Fig. A.5. We use CIFAR10, 10 clients with full participation, $\alpha = 0.1$, and $\rho \cdot n_i = 4500$. We observe that when $p = 0.06$, though the stochastic noise $\hat{\sigma}_p^2$ remains similar, $\hat{\zeta}_p^2$ has reduced by half and we improve the Top-1 accuracy by 20%. When $p = 0.5$, we obtain a much smaller $\hat{\sigma}_p^2$ and $\hat{\zeta}_p^2$. Consequently, we achieve an even better Top-1 accuracy than the IID experiment. However, the parameters in Fig. A.5 (b) and (c) are evaluated with different server models $\mathbf{x}_{p,r}$, so it is less comparable to Lemma I where they are measured with the same server model. Therefore, we evaluate the gradient dissimilarity $\hat{\zeta}_p^2$ and stochastic noise $\hat{\sigma}_p^2$ using $\mathcal{D}_i \cup \tilde{\mathcal{D}}_{si}$ and the corresponding $\sigma^2$ and $\zeta^2$ using $\mathcal{D}_i$ with the same server model $\mathbf{x}_r$ in each round in Fig. A.5 (d). We observe that $(1-p)\sigma^2$ dominates over other terms in the effective stochastic noise, which means the first term in the convergence rate for non-convex function in Corollary I can be simplified as $\mathcal{O}\left(\frac{(1-p)\sigma_p^2 L_p}{n\varepsilon^2}\right)$ in this experimental setup. For $\hat{\zeta}_p^2$, the empirical result also matches the theory.

These results show that in this experimental setup, adding shuffled synthetic dataset reduces both stochastic noise and function dissimilarity, and lead to a greater accuracy improvement.

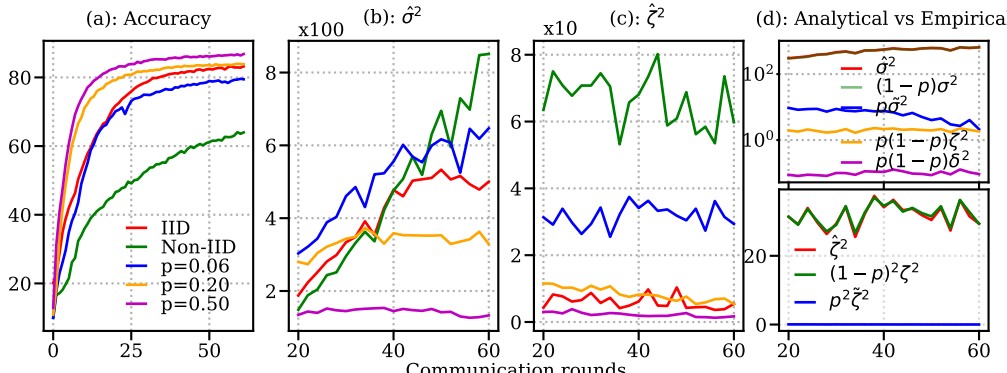

Figure A.5: Influence of using shuffled synthetic dataset on the stochastic noise $\hat{\sigma}^2$ and function dissimilarity $\hat{\zeta}^2$ (CIFAR10, 10 clients, $\alpha = 0.1$). The percentage of shuffled data in $(d)$ is 0.06. The empirical observation of $\hat{\sigma}^2$ and $\hat{\zeta}^2$ matches the theoretical statements.

### A.4  EXPERIMENTAL RESULTS ON DSPRITES DATASET

In this section, we demonstrate the effectiveness of our proposed framework on the dSprites dataset Matthey et al. (2017). We here perform a slightly different synthetic data generation process. Rather than using the DDPM Ho et al. (2020), we generate the synthetic dataset on each client with $\beta$-VAE Burgess et al. (2018). We then aggregate and shuffle the collected synthetic dataset on the server and distribute them to clients following the same procedure as described in Sec. 3 and Sec. 4.

### A.4.1  TRAINING DETAILS

The dSprites dataset Matthey et al. (2017) contains three different types of shapes with different colours, scales, rotations, and locations.

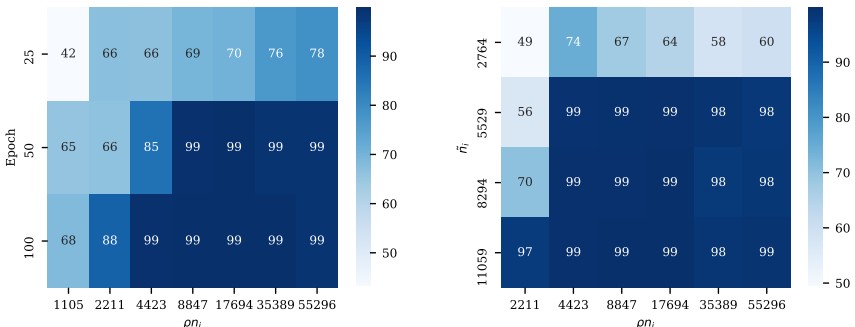

Figure A.6: Performance on the dSprites test dataset. Left: the influence of the number of training epochs and training images ($\rho n_i$) in training the generator on the FL performance. Right: the influence of the number of synthetic images ($\tilde{n}_i$) on the FL performance. When we use fewer training images to train the generator (e.g. 2211), it is beneficial to train the generator longer and sample more synthetic images.

**Data preparation** We first randomly select 10% of the images from the dSprites dataset to formulate the test set. We leave the test set on the server to evaluate the performance of the server model. For the rest of the dataset, we split them among 12 clients ($N = 12$) based on the four spatial locations (top-left, bottom-left, top-right, or bottom-right) and three shapes (square, ellipse, or heart) such that each client only sees a single type of shape from one of the four pre-defined locations. The number of images on each client is the same ($n_i = 55296$). We consider each shape as a class and perform shape classification.

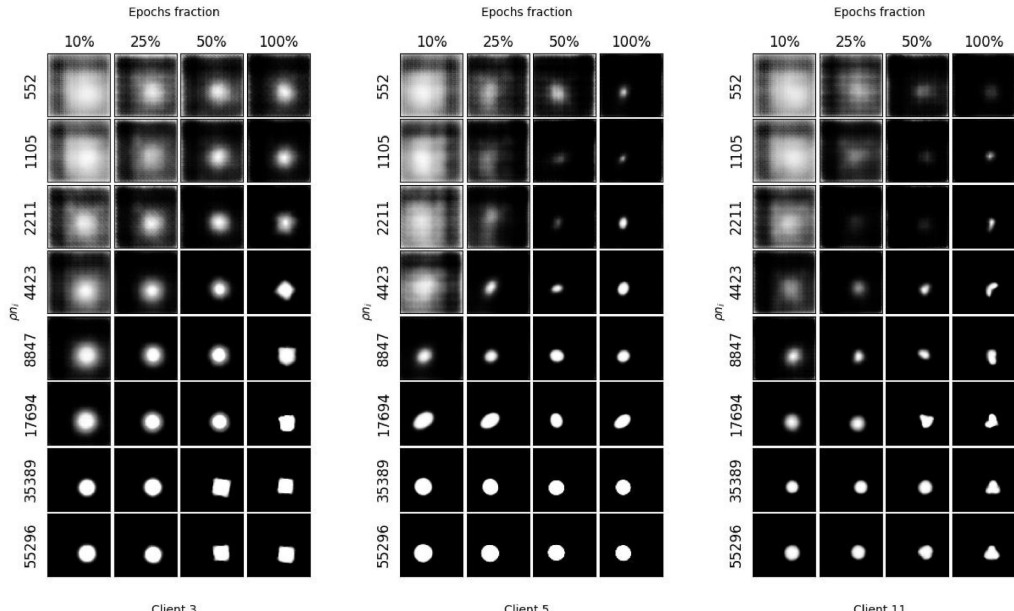

Figure A.7: Example synthetic images from clients 3, 5, and 11. The x-axis shows the training fraction of 360 epochs, and the y-axis shows the number of training images used to train the generator. The number of training images has a more substantial influence on the quality of the synthetic data.

**Synthetic data generation** We use $\beta$-VAE with the same architecture as Burgess et al. (2018) following a publicly available implementation We train an individual $\beta$-VAE Burgess et al. (2018) on each client with batch size 256, latent dimension 10. To evaluate the sensitivity of the federated learning optimization on the quality of the generator, we train the generator using different fractions of data $\rho n_i$ with $\rho$ being $\{1\%, 2\%, 4\%, 8\%, 16\%, 32\%, 64\%, 100\%\}$ on each client. When $\rho = 100\%$, we train a $\beta$-VAE with the entire dataset from each client.

Many works have demonstrated that the latent variables in $\beta$-VAE can encode disentangled representative features of the input images Burgess et al. (2018); Matthey et al. (2017); Kingma & Welling (2013), and manipulating a single latent dimension can result in substantial changes of the corresponding factor of variation, e.g., scale, in the output from the decoder. Therefore, we can obtain diverse synthetic images by interpolating the extracted latent representations. To achieve this, we extract the averaged latent representations for each class, specifically the mean of the posterior distribution $\mu_c$ Burgess et al. (2018). We then sample $\tilde{n}_i$ latent codes from the Gaussian distribution parameterised by mean $\mu_c$ and standard deviation 1. The sampled latent codes are then passed as the input for the decoder from $\beta$-VAE to generate synthetic images. We collect the synthetic images from all the clients, which are then shuffled and distributed equally to each client such that the local dataset on each client is $p\mathcal{D}_i + (1 - p)\tilde{\mathcal{D}}_{si}$. An example of the generated synthetic image is shown in Fig. A.7. With the updated dataset on each client, we follow the same procedure as documented in Sec. 4 to perform federated optimization with FedAvg McMahan et al. (2016).

### A.4.2 DISCUSSION

We show the sensitivity of FedAvg's performance on the quality of the synthetic images in Fig. A.6. The left image shows that training the VAE longer does not necessarily provide better quality synthetic images when we use fewer training images. However, when we use more training images (higher $\rho$), the images extracted from early checkpoints, e.g., 50% epochs, are already of high quality. We observe a similar pattern in the right image in Fig. A.6, where the number of training images has a more substantial influence on the performance of FedAvg than the number of generated synthetic images.

