# OpenReview forum: "Synthetic data shuffling accelerates the convergence of federated learning under data heterogeneity"
_ICLR.cc/2024/Conference — ICLR 2024 Conference Withdrawn Submission_

### Official Review · Reviewer_zeqb · 2023-10-29

**Soundness:** 2 fair
**Presentation:** 2 fair
**Contribution:** 2 fair
**Rating:** 5
**Confidence:** 3

**Summary:**

This paper proposes to use shuffled synthetic data to combat the heterogeneity issue in federated learning (FL). Specifically, it proposes to train a local generator for each client, generate a synthetic dataset from the trained generator, and then send it to the server. The server then aggregates and shuffles the synthetic data from all the clients and sends a uniform partition to each client. Each client then does its local training on a combination of its original local dataset and the shuffled synthetic data received from the server. The effectiveness of the proposed method is shown via several experiments. There is also some theoretical analysis to quantify the effect of data shuffling on the convergence of FL.

**Strengths:**

Synthetic data has shown a lot of promise in machine learning problems. This paper shows its effectiveness when properly shuffled, in improving the convergence of several federated optimization algorithms. The empirical gains shown are significant in many cases. It seems that this paper explains how it is better than prior work on the usage of synthetic data in federated learning, but I'm not very knowledgeable about this particular area or familiar with related works.

**Weaknesses:**

**1.** In the discussion around Lemma 1 and Corollary I, it is mentioned that the asymptotic convergence mainly depends on $\hat{\sigma}_p^2$. From the expression of $\mathbb{E}[\hat{\sigma}_p^2]$ in Lemma 1, one can compute that $\frac{d \text{ } \mathbb{E}[\hat{\sigma}_p^2]}{d p} = (1-2p)(\zeta^2 + \delta^2) - (\sigma^2 - \tilde{\sigma}^2)$. Now, for non-iid situations when $\zeta^2 \gg (\sigma^2 - \tilde{\sigma}^2)$, $\mathbb{E}[\hat{\sigma}_p^2]$ is an *increasing function* of $p$ around $p=0$ (no synthetic data). So, the convergence should get **worse** according to this result with small values of $p$ or a little bit of synthetic data.

In fact, I think the value of $p$, say $p^{\ast}$, such that $\mathbb{E}[\hat{\sigma}_{p^{\ast}}^2] < {\sigma}^2$ is going to be large (this can easily be calculated because it is quadratic in $p^{\ast}$).

Can the authors address this?

**2.** I don't understand the second-last sentence on page 4 "*When $\sigma^2=0$, or in general...*". Can you quantify "super-linearly"? Also, where is $\sqrt{L_p}$ coming up in the bound of Corollary I? Overall this statement doesn't make too much sense to me.

**3.** The paper talks about privacy of the local data but there are no formal privacy guarantees given with the proposed method. Moreover, training a generator on each client is also expensive.

**Questions:**

Please see Weaknesses above. If W1 and W2 are addressed well, I can raise my score.

---

### Official Review · Reviewer_wuSQ · 2023-11-01

**Soundness:** 2 fair
**Presentation:** 3 good
**Contribution:** 1 poor
**Rating:** 3
**Confidence:** 5

**Summary:**

This paper studies the problem of federated learning using Local SGD. The paper assumes there are $N$ clients with heterogeneous data and they want to minimize the average objective. This is the standard in federated learning, but the paper diverges from prior work by considering a scenario where the clients share a fraction $p$ of their data. They show (Lemma 1, Corollary 1) that this (naturally) leads to a lower variance term, and hence faster convergence  (as shown in their Figure 2). Because sharing data directly contradicts user privacy (a cornerstone of federated learning), the paper proposes to train a generator on each client and send synthetically generated data instead. If the generator is good enough, then the results of (Lemma 1, Corollary 1) apply.

**Strengths:**

1. The paper is well-written and clear.
2. The presented theory is correct and sound as far as I can see.

**Weaknesses:**

1. It is already known that reshuffling data locally reduces the variance (see e.g. [1]). Why is it any harder to study this problem across clients?
2. This is the main dealbreaker: sending synthetic images still breaks privacy. Even on massive datasets, generator models routinely reproduce training data (see [2, 3]). The data on any client in an ordinary federated learning problem is several orders of magnitude smaller than ImageNet or other large training sets, it is virtually guaranteed that a synthetic generator trained on local data would simply regurgitate client data.

My opinion is that the paper's results are not very theoretically surprising, and its practicality is nil because of privacy considerations. For this reason, I recommend rejection.

[1] Mishchenko, Khaled and Richtárik. Proximal and Federated Random Reshuffling, ICML 2022.
[2] Tinsley, Czajka, and Flynn. "This face does not exist... but it might be yours! identity leakage in generative models." IEEE/CVF Winter Conference on Applications of Computer Vision. 2021.
[3] Carlini N, Hayes J, Nasr M, Jagielski M, Sehwag V, Tramer F, Balle B, Ippolito D, Wallace E. Extracting training data from diffusion models. In32nd USENIX Security Symposium (USENIX Security 23) 2023 (pp. 5253-5270).

**Questions:**

N/A.

---

> ### Author Response · Authors · 2023-11-21
>
> We thank the reviewer for the evaluation of our paper. Your every comment is important to us. We provide answers to your concerns below:
>
> > W1. It is already known that reshuffling data locally reduces the variance (see e.g. [1]). Why is it any harder to study this problem across clients?
>
>
> We thank the reviewer for the reference. Shuffling data locally does not change the data heterogeneity across clients, which is one of the most critical issues in FL and the problem we address in this paper. We have provided explicit formulas regarding how shuffling data across clients influences gradient dissimilarity and accelerates convergence.
>
>
> > W2. sending synthetic images still breaks privacy.
>
> We totally agree with the reviewer regarding the privacy implication of the proposed approach, as we have also acknowledged the limitations of our approach in the paper.
>
> - Sharing the synthetic data can indeed give the server a chance to infer the local data distribution. However, the clients have complete control over what they share.
> - Besides, even without sharing synthetic data (vanilla FL algorithm, such as FedAvg), malicious servers can still perform inference attacks to predict the client's training dataset [Qian et al. 2021][Melis et al. 2018] based on the model update.
> - Additionally, we would like to draw your attention to Table A.2, which illustrates a significant reduction in communication costs, particularly in heterogeneous data scenarios. This reduction highlights the practical advantages of our approach.
>
> Therefore, we respectfully disagree with the reviewer's notion that our approach's practicality is negligible.
>
> Qian et al. 2021, What can we learn from gradients?
>
> Melis et al. 2018, Exploiting Unintended Feature Leakage in Collaborative Learning.

---

> > ### Comment · Reviewer_wuSQ · 2023-11-21
> >
> > > We thank the reviewer for the reference. Shuffling data locally does not change the data heterogeneity across clients, which is one of the most critical issues in FL and the problem we address in this paper. We have provided explicit formulas regarding how shuffling data across clients influences gradient dissimilarity and accelerates convergence.
> >
> > Shuffling data across clients is theoretically the same problem as shuffling data within clients, shuffling data within clients reduces local statistical estimation variance while shuffling data across clients reduces across-client statistical estimation variance. From the perspective of theory, the difference between both things is minor, and therefore I'm not really sure what new theoretical insight is here.
> >
> > > We totally agree with the reviewer regarding the privacy implication of the proposed approach, as we have also acknowledged the limitations of our approach in the paper.
> > > Sharing the synthetic data can indeed give the server a chance to infer the local data distribution. However, the clients have complete control over what they share.
> >
> > The idea that every client will look at the generated synthetic data before sending it is very impractical. Take Google Keyboard, for instance, the whole premise there is that the training is done in the background and with assurances to the client that their data never leaves their phone. This approach (a) breaks this assurance, and (b) requires clients to perform *more* work.
> >
> > > Besides, even without sharing synthetic data (vanilla FL algorithm, such as FedAvg), malicious servers can still perform inference attacks to predict the client's training dataset [Qian et al. 2021][Melis et al. 2018] based on the model update.
> >
> > While this is true, there is plenty of literature on using approaches from differential privacy to avoid any such attacks. This literature does not apply here, because you're simply sending data and not model updates.
> >
> > > Additionally, we would like to draw your attention to Table A.2, which illustrates a significant reduction in communication costs, particularly in heterogeneous data scenarios. This reduction highlights the practical advantages of our approach.
> >
> > If you simply sent all local data to the server, the communication cost would be close to zero because then the server can just use this data to minimize its local objective. A reduction in communication cost means very little if the approach breaks one of the central tenets of federated learning (all data stays local).

---

> > > ### Author Response · Authors · 2023-11-21
> > >
> > > We thank the reviewer for the fast reply and help in improving the paper. We will enhance the practicality of our approach by making the generator differentially private in our future work.

---

### Official Review · Reviewer_suLd · 2023-11-03

**Soundness:** 3 good
**Presentation:** 4 excellent
**Contribution:** 2 fair
**Rating:** 5
**Confidence:** 4

**Summary:**

This paper presents a framework for accelerating federated learning based on synthetic data. The framework first trains generators on each client, then sends the generated samples to the server. The server aggregates the synthetic samples and distributes the generated samples to clients in i.i.d. manner. Authors provided rigorous analysis to their proposed method and provided extensive experiment evidence to show synthetic data improves both convergence and test accuracy.

Without differential privacy (DP), I have some concerns on whether the proposed method goes against the motivation of federated learning in the first place. For example, Large Language Models are shown to be capable of memorizing parts of the training set:

Quantifying Memorization Across Neural Language Models, Carlini et al. (ICLR 2023)

With that said, the current setup of the experiments provided quantitative evidence of the benefit of synthetic data, although the idea of generating synthetic data to enhance federated learning is not novel.

**Strengths:**

The paper is well-written. The theoretical analysis is easy to follow. Authors provided extensive ablation of their proposed method.

**Weaknesses:**

1. This paper does not address important privacy concerns of transmitting synthetic samples generated on clients.

2. To evaluate the paper by leaving privacy concerns out of the scope, there are a few aspects I would like to obtain more insights:


       i. if not for privacy concerns, would FL still add value on top on synthetic data training obtained via stronger generative models fine-tuned on clients?  For example, by using state-of-art diffusion model for image, or Large language model for NLP problems.


     ii. I would like to see an ablation on the synthetic data generator used. Different generator quality should theoretically correspond to different $\delta$ in Assumption 4. Hence I would like to see how $\delta$ influence the quality backed by experiments. This will also enhance the practicality and shed some insight on the tightness of Corollary 1.

**Questions:**

Corollary I cites the framework developed in

"A unified theory of decentralized SGD with changing topology and local updates. "

If I am correct, this establishes the convergence for distributed optimization on the aggregated dataset $(1-p)\mathcal{D}_i + p\mathcal{D}$. This provides little value to understand the convergence aspect since the quantity of interest are defined on the true data distribution which is union of $\mathcal{D}_i $.


Corollary I does not explicitly defined the quantity it tries to bound, hence my understanding can be wrong and please clarify.

---

### Official Review · Reviewer_s3fS · 2023-11-06

**Soundness:** 2 fair
**Presentation:** 3 good
**Contribution:** 2 fair
**Rating:** 5
**Confidence:** 4

**Summary:**

This work revolves around the notion of reshuffling data across clients (as a preprocessing subroutine on top of established federated methods) in the Federated Learning setting aiming to homogenize their local distributions thus deriving improved accuracy and convergence speed. Specifically each client generates synthetic data based on their local distribution which are sent to the server. The server shuffles the received data and transmit them back equally to the clients thus augmenting their initial dataset. Theoretical justification is provided suggesting that the convergence speed is enhanced through this mechanism and experimental result on Cifar10 and Cifar100 that support the theoretical findings are presented in the main body of the paper.

**Strengths:**

-The paper is well-structured and easy to follow.

-The theoretical justification provided is intuitive and contributes to better understand the effects of data heterogeneity on the convergence speeds of the federated algorithms.

-Experimental results support the main point of the paper and suggest that there are settings where the proposed mechanism can alleviate the issue of data heterogeneity in the framework of FL.

**Weaknesses:**

-The theoretical results although insightful, appear to be straightforward derivations applied to known convergence bounds (Koloskova et al., 2020; Woodworth et al., 2020; Khaled et al., 2020). Furthermore, the idea of data reshuffling in FL is well known and as a result the theoretical contribution and the novelty of this work is somewhat marginal.

-A major concern lies on the metric of accuracy that is used in this work. Specifically, in the proposed method the clients obtain an augmented dataset which could potentially be very different from their initial data. As a result, achieving high accuracy on these augmented datasets is a potentially substantially easier task which however is misaligned with the true objectives of the clients. The authors need to further discuss and clarify this issue. Specifically, in the experiments presented it is my understanding that the size of the initial dataset for client $i$, is equal in size to the synthetic datasets that is assigned to the same client $n_i = \tilde{n}$. If indeed this is the case the resulting accuracy could be a misleading metric of success.

-This method relies on clients being able to produce synthetic data of good quality. In many FL settings however clients often have access to very few samples which could significantly hinder the effectiveness/soundness of the proposed method.

-In the comparison with other synthetic-data based approached it would be beneficial to include convergence plot instead of just providing a final accuracy table. Further, the parameters for DENSE should be chosen optimally to provide a fair comparison.

-The comment on page 9 in section "Practical implication" claiming that "Additionally, the client has the option of checking the synthetic images and only sharing the less sensitive synthetic images to alleviate the information leakage." seems unreasonable. It is hard to imagine clients choosing manually images that diminish the information leakage out of a big number of synthetically produced data.


-Minor issues

- Page 3, end of paragraph 1: "fail achieve" --> "fail to achieve".
- Page 4, paragraph 3: "it might infeasible" -->"it might be infeasible"

**Questions:**

-A major concern lies on the metric of accuracy that is used in this work. Specifically, in the proposed method the clients obtain an augmented dataset which could potentially be very different from their initial data. As a result, achieving high accuracy on these augmented datasets is a potentially substantially easier task which however is misaligned with the true objectives of the clients. The authors need to further discuss and clarify this issue. Specifically, in the experiments presented it is my understanding that the size of the initial dataset for client $i$, is equal in size to the synthetic datasets that is assigned to the same client $n_i = \tilde{n}$. If indeed this is the case the resulting accuracy could be a misleading metric of success.

---

> ### Author Response · Authors · 2023-11-21
>
> We thank the reviewer for the constructive and helpful comments. Your every comment is important to us. You can find our replies below:
>
> > W2. A major concern lies on the metric of accuracy that is used in this work.
>
> We apologize for not making this clear.
> Our reported test accuracy is always evaluated on the real server test dataset. No synthetic data is involved in calculating the test accuracy.
>
>
> > W3. This method relies on clients being able to produce synthetic data of good quality. In many FL settings however clients often have access to very few samples which could significantly hinder the effectiveness/soundness of the proposed method.
>
> We agree with the reviewer that better test accuracy can always be obtained with better-quality synthetic data. However, even if the quality of the synthetic data is bad, we can still get a better test accuracy compared to the vanilla federated learning algorithm (see Fig.4 (b) when E=50 and the corresponding synthetic images in Fig A.2 bottom left)
>
> > W4. In the comparison with other synthetic-data based approached it would be beneficial to include convergence plot instead of just providing a final accuracy table. Further, the parameters for DENSE should be chosen optimally to provide a fair comparison.
>
> We agree with the reviewer that we can potentially improve the performance of DENSE if we tune the hyperparameter more exhaustively and carefully.
>
> As the reviewer requested, we have provided the test accuracy curve of FedGEN in Fig. A.1.
>
>
> > W5. The comment on page 9 in section "Practical implication" claiming that "Additionally, the client has the option of checking the synthetic images and only sharing the less sensitive synthetic images to alleviate the information leakage." seems unreasonable. It is hard to imagine clients choosing manually images that diminish the information leakage out of a big number of synthetically produced data.
>
> We apologize for not making this clear. We agree that if the number of synthetic images is large, it can be time-consuming to filter out those sensitive synthetic data points manually. One possible way to address this is to automate this process with methods such as the Siamese network [Koch et al. 2015] locally.
>
> Koch et al. 2015 Siamese Neural Networks for One-shot Image Recognition
>
>
> > W5. minor issues
>
> We thank the reviewer for pointing them out and have fixed them in the revised manuscript.

---

### Author Response · Authors · 2023-11-21

We thank all the reviewers for their valuable and insightful feedback. We are glad that the reviewers found the paper well-written (**s3fS, suLd, wuSQ**), agreed that we have provided a theoretical justification for the effectiveness of the proposed approach (**s3fS, suLd**), acknowledged our extensive ablation study (**suLd**), and the theory is correct and sound (**wuSQ**).

One of the main concerns several reviewers raised is the potential privacy issue while sharing the local synthetic data. We agree on this. However, due to the time limit, we cannot perform extra experiments. Therefore, we decided to withdraw our submission.